

# Tropical Moisture Exports, Extreme Precipitation and Floods in Northeast US

Mengqian Lu[1,2], Upmanu Lall[2,3]

[1]Department of Civil & Environmental Engineering, Hong Kong University of Science and Technology, Kowloon, Hong Kong
[2]Columbia Water Center, Columbia University, New York, NY, USA
[3]Department of Earth & Environmental Engineering, Columbia University, New York, NY, USA

*Correspondence to*: Mengqian Lu (mengqian.lu@ust.hk)

**Abstract.** A statistically and physically based framework is put forward that investigates the relationship between Tropical Moisture Exports (TME), and extreme Precipitation and floods in the Northeast United States (N.E. USA). TME correspond to the meridional transport of moist air masses, primarily born in tropical oceanic areas, to higher latitudes; contribute to the global climatology precipitation and its extremes; and are closely related to flood events, especially in the mid-latitudes. The birth process and the steering of TME have seasonal and interannual variability. In this study, we explore how the TME are related to extreme precipitation and floods in the N.E. USA with a focus on seasonal variability and the potential impact of the El Niño Southern Oscillation. Links of TME to large floods events in N.E. USA in different seasons are first identified. The major moisture sources of the TME that contribute to precipitation extremes and floods in N.E. USA are then identified, together with the seasonally and interannually varying characteristics in terms of both TME birth and entrance to the N.E. USA, and their subsequent contribution to extreme precipitation. We show that the extreme daily precipitation events are dominated by extreme TME entering the N.E. USA events in every season.

## 1 Introduction

Surface temperature gradients (*Lorenz*, 1984*; Jain et al.,* 1999*; Karamperidou et al.,* 2012) drive the large scale atmospheric circulation, and hence the meridional transport of moist air masses, primarily born in tropical oceanic areas, to higher latitudes, often manifested as localized threads of strong moist air fluxes in the warm sector of extratropical cyclones. There are a number of studies focusing on





'Atmospheric Rivers' (ARs), a concept which was first introduced by (*Zhu and Newell*, 1994) who defined an atmospheric phenomenon that features a narrow corridor of concentrated moisture, enhanced water vapor transport that have a large hydrologic effect. Recent studies have linked Atmospheric Rivers *(Zhu and Newell,* 1998*; Bao et al.,* 2006) to extreme precipitation and floods (*Ralph et al.,* 2006*;*

*Leung and Qian,* 2009*; Lavers et al.,* 2011, 2013*; Ralph and Dettinger,* 2011*; Lu et al.,* 2013*; Lavers et al.,* 2016) as the major hydrometeorological contributor. ARs have been linked to extreme precipitation in the United Kingdom (*Lavers et al.,* 2011, 2013*; Nakamura et al.,* 2013), Western France (*Lu et al.*, 2013),  Midwest United States (*Nakamura et al.*, 2013) and the West coast of United States (*Ralph et al.,* 2006*; Dettinger,* 2011*; Ryoo et al.,* 2011*; Lavers et al.,* 2016) which has been linked to the well-

known "Pineapple express" (*Higgins et al.,* 2000), that originates from oceanic areas adjacent to Hawaiian Islands. However, areas such as the Northeast United States (N.E. USA) have not been studied as much in this context.

The widely used definition of ARs as a narrow plume with at least 2cm of integrated water vapor

(IWV), extending over at least 2000 km long and 1000 km wide (*Ralph et al.,* 2004*; Neiman et al.,* 2008*; Dacre et al.,* 2015), has restricted the studies to popular AR-associated regions as indicated in the preceding paragraph. In contrast, *Bao et al.* (2006) and *Wernli* (1997), suggested the term "moist conveyor belt" to broaden the scope of the studies to Tropical Moisture Exports (TME) that may have significant contributions to hydrometerological extremes on a global scale *(Knippertz and Wernli,* 2010*;*

*Knippertz et al.,* 2013*; Nakamura et al.,* 2013*; Lu et al.,* 2013). Here, we explore how TME may be involved in moisture transport into the N.E. USA, and how their sources and influence vary by season and under different ENSO regimes. The linkage of TME to extreme precipitation and flooding in each of the seasons is also assessed.

Past studies on the nexus of moisture transport, extreme precipitation and floods, focused on the (1) link to selected historical floods events in various regions (*Ralph et al.*, 2006; *Lavers et al.*, 2011; *Lu et al.*, 2013; *Nakamura et al.*, 2013); (2) identification of moisture sources that contributes to extremes (*Knippertz and Wernli*, 2010; *Ryoo et al.*, 2011; *Knippertz et al.*, 2013; *Lu et al.*, 2013; *Nakamura et al.*,




2013); and (3) trajectory analysis of the air masses with investigation of the attendant atmospheric circulation (*Wernli*, 1997; *Bao et al.*, 2006; *Ryoo et al.*, 2011; *Nakamura et al., 2013; Lu et al.*, 2013). *Lavers et al.* (2011) linked ARs with the top 10 largest winter floods in Britain since 1970; while *Nakamura et al.* (2013) identified strong association between anomalous atmospheric circulations that

drive low-level flow of warm and moist air, and 21 extreme floods in Ohio River. Further, *Knippertz and Wernli* (2010), *Lu et al.* (2013) and *Nakamura et al.* (2013) note that the poleward transport of tropical born moist air masses to the Northern Hemispheric extratropics provide the link between tropical moisture sources and extratropical extreme precipitation, occasionally with explosive cyclogenesis. *Lu et al.* (2013) associated TME from the Gulf of Mexico and Tropical North Atlantic

Ocean (TNAO) east to the Bahamas islands as the major moisture sources for the 1995 January flood in western France, and demonstrated the predictability of the extreme precipitation given only the mid-latitude sea level pressure (SLP) fields, suggesting that steering mechanisms were important. A similar study for the extreme floods in Ohio River Basin by *Nakamura et al.* (2013) identified a persistent dipole pattern in the SLP leading to the wave like transport of moisture from the Gulf of Mexico into

the flooded area, every 4 to 7 days over the March-April-May season.

## 1.1 Tropical Moisture Export Characterization

Tropical Moisture Exports (TME) was first documented in (*Knippertz and Wernli*, 2010) for Northern Hemisphere and later extended by *Knippertz et al.* (2013) to produce a global TME climatology. The

TME tracks were calculated using 6-hourly ERA-Interim data (*Dee et al.*, 2011) and range from 1989 to 2010, covering daily tracks born in the tropics $[0° - 20°N]$. Each trajectory has its moisture source calculated for every 100 km × 100 km box between the equator and 20°N, and for every 30hPa between 1000 and 490hPa, such that 90% of all water vapor is integrated. Each trajectory represents $3 × 10^{12}$ kg of atmospheric mass. Trajectories are calculated with the LAGRANTO Lagrangian analysis tool

(*Wernli*, 1997; *Wernli and Davies*, 1997), by interpolating the relevant fields to the positions of the trajectory at 6-hr updating frequency. To ensure that the characteristics of the tropical air parcels are maintained on their way across the subtropics, only trajectories that reach 35°N within the next 5 to 6





days after crossing 20°N were retained; however, changes due to fluxes of heat and moisture from the underlying surface or mixing cannot be completely excluded. The water vapor fluxes of the retained tracks in the dataset must reach 100 g Kg$^{-1}$ m s$^{-1}$, a threshold chosen to represent 'fast' events and yet get meaningful statistics (*Knippertz and Wernli*, 2010). *Knippertz and Wernli* (2010) showed that TME

contributes significantly (more than 60% of the average) to climatological precipitation in the mid-latitudes and identified four major source regions in the tropics:

(1) "Pineapple Express' (PE) [170° – 130°W]. This source has a maximum activity in the DJF season and is almost absent in JJA.

(2) 'Great Plain' (GP) [100° – 90°W]. We extended this region to [100° – 70°W] after initial analyses

revealed that it was the dominant source region for the TME entering the N.E. USA. This region includes the Gulf of Mexico, and also parts of the continental region between the Rocky and the Appalachian Mountains.

(3) 'Gulf Stream' (GS) [40° – 70°W]. This source is active year round with low seasonality.

(4) 'West Pacific' (WP) [120° – 170°E]. This source has been associated with the mei-yu-baiu front

over East Asia with activity from MAM to SON peaking in JJA.

Subsequently, we identify the TME tracks as PE, GP, GS or WP as associated with their birthplace.

The conceptual framework of the analysis presented in this paper is indicated in Figure 1. The causal structure illustrated considers the potential dependence of the TME Birth process as a function of the

source location, the season and ENSO state. The number of TME that enters the N.E. USA on any given day depends on the associated birth process, the season, the source, the ENSO state, and the atmospheric circulation. The total water released ($\Delta Q$) by the TME in the N.E. USA on a given day is taken to depend on the number of TME entering. The extreme precipitation amount, EP, is considered to depend on the $\Delta Q$. We take the N.E. USA [39°N – 48°N, 66°W – 82°W] as the study area to address

the following questions:

(1) How were floods events in N.E. USA in different seasons related to TME: the moisture sources and their seasonality?

(2) What are the birth mechanism of TME and the role of large scale climate regulation?



(3) What are the entrance mechanism of TME to N.E. USA and the role of identifiable Atmospheric circulation patterns?

(4) What is the link between TME and extreme precipitation events that may trigger floods?

The paper is organized in line with answering these research questions. The data used in this study is
5 provided in Section 2. In Section 3, we illustrate the association of TME with examples of extreme floods in each season. The second research question is addressed in Section 4. The entrance of TME to the N.E. USA is discussed in Section 5. The link between moisture release and extreme precipitation that is closely related to floods is explored in Section 6. A discussion and summary of the key findings is provided in Section 7.

**2 Data**

We use the TME dataset documented in (*Knippertz and Wernli*, 2010). The dataset covers from 1989 to 2010, recording daily tracks born in the tropics that meet the following criteria: (1) they reach 35°N within the next 5 to 6 days after crossing 20°N, and (2) water vapor flux of any track is not less than 100 g Kg$^{-1}$ m s$^{-1}$. The position of the air parcel was updated every 6 hours, thus each track has 29 (4 updates
up to 7 days including birth place, 4×7+1) positions (latitudes & longitudes) recorded on its trajectory.

The change of moisture (recharge or release) along each track is calculated as

$$\Delta Q_k(j) = Q_k(j) - Q_k(j+1),\qquad (1)$$

where $j$ is the time point on the trajectory (the $j^{th}$ position along the track), $Q_k(j)$ is the $k^{th}$ TME track's
specific humidity in g Kg$^{-1}$ (units) at time point $j$ and $\Delta Q_k(j)$ is the change of specific humidity. Each of the tracks has its positions recorded every 6 hours, up to 7 days, with a total of 29 time points. Thus, $j$ ranges from 0 to 28, with the birth location at $j = 0$ and the point before death location at $j = 28$.




The total release of water from the air parcel in the study area was calculated as the change of total specific humidify (including liquid phase) integrated over all the tracks for a given date starting from their entrance as follows:

$$\Delta Q(t) = \sum_{k=1}^{N_t} \left( Q_k(t) - Q_k(t+1) \right), \qquad (2)$$

where $\Delta Q(t)$ is the total change of specific humidity of all the tracks that are active in the N.E. USA on day $t$, $N_t$ is the total number of tracks active in the N.E. USA on day $t$. Note that we consider the tracks leaving/exiting at the different hours of the day, i.e. 0 o'clock, 6 o'clock, 12 o'clock, 18 o'clock, 24 o'clock. By examining the selected tracks and their release of moisture to the study area, the associated moisture birth location and their trajectories to the N.E. USA can be identified and the

precipitation resulted from the release of water vapor from the moist air parcel can also be computed.

For the analysis of atmospheric circulation patterns, we choose the sea level pressure (SLP) data from the NCEP/NCAR Reanalysis project (*Kalnay et al.*, 1996) provided by the NOAA/OAR/ESRL PSD, Boulder, Colorado, USA, from their Web site at http://www.esrl.noaa.gov/psd/ . We use the daily SLP data, with a resolution of 2.5° latitude× 2.5° longitude, covering the same period, 1989 – 2010. We

further derived the daily SLP anomalies against calendar day climatology as calculated as follows.

$$SLPa_{ij} = SLP_{ij} - SLPc_i$$

$$SLPc_i = \frac{\sum_{j=1989}^{2010} SLP_{ij}}{22} \qquad (3)$$

where $i$ is the $i^{th}$ day in year $j$, $i = 1, \ldots, 365$ (366 for leap years) and $j = 1989, \ldots, 2010$; $SLPa_{ij}$ is the SLP anomaly on $i^{th}$ day in year $j$, $SLPc_i$ is the daily climatology of SLP on the $i^{th}$ day of a year, or average calendar day SLP.

The Oceanic Niño Index (ONI) is provided by NOAA/National Weather Service, NOAA Center for Weather and Climate Prediction, Climate Prediction Center, from their website at http://www.cpc.ncep.noaa.gov/. The ONI has become the de-facto standard that NOAA uses for identifying El Niño (warm) and La Niña (cool) events in the tropical Pacific. The warm and cold



episodes are based on a threshold of $\pm 0.5^{\circ}$C of the running 3-month mean of ERSSTv3b data (*Smith et al.*, 2008) SST anomaly for the Niño 3.4 region (i.e., 5°N – 5°S, 120° – 170°W). Cold and warm episodes are defined when the threshold is met for a minimum of 5 consecutive over-lapping seasons.

The flood events in N.E. USA we discuss in the following section are recorded by the Dartmouth Flood Observatory (DFO, at their website: http://floodobservatory.colorado.edu/). The Dartmouth Flood Observatory Achieve contains global floods events from 1985 to present based on data acquired by NASA, the Japanese Space Agency and the European Space Agency.

**3 TME and Floods in the N.E. USA**

For each of the four seasons, i.e., Dec – Jan – Feb (DJF), Mar – Apr – May (MAM), Jun – Jul – Aug (JJA) and Sep – Oct – Nov (SON), major flood events in the N.E. USA as recorded in the DFO data base were identified. For each event, TME tracks that were born in any of the four source regions within 7 days of the onset of heavy precipitation in the flooded region were identified. Of these, those that entered the N.E. USA, including those that continued out of the region, were identified. A

representative flood event is identified for each season and the TME tracks associated with it are shown in Figure 2. The colors indicate the changes of specific humidity along the tracks, calculated as in Eq.(1). Moisture release is recorded as blue or light blue dots in Figure 2, while moisture recharge magnitude is shown using red or yellow dots. Basic attributes of the exemplified flood events are summarized in Table 1.

Figure 2 (a) – (f) shows that the associated TME for the January 1996 flood (Table 1) were born in (1) 'Pineapple Express' (PE), i.e. Niño 3.4 region near Hawaii; (2) 'Great Plain' (GP), i.e. Gulf of Mexico and (3) 'Gulf Stream,' (GS), i.e. Tropical North Atlantic Ocean (TNAO) east to the Bahamas. The beginning of the heavy rainfall appears related to tracks born in all the three regions as shown in Figure

2 (a) – (c). Later in the event (Figure 2 (d) – (f)) TME that entered the N.E. USA were all born in GP and GS. The change of moisture content of the tracks indicates that for tracks born in PE, the relatively longer travel time and distance resulted in more recharging and releasing on their way to the N.E. USA,



compared to GP and GS, which featured less changes of moisture content before reaching the flooded area, and relatively more release in the N.E. USA (more dark blue dots of tracks in Figure 2 (d) – (f) in the red box – N.E. USA). There are strong spatial patterns of the trajectories given the TME tracks' sources: tracks from PE follow typical Northeastward 'Pineapple Express' sine wave like trajectory

5    widely showed in the AR literature [e.g., (*Dettinger*, 2011; *Dettinger et al.*, 2011; *Ralph and Dettinger*, 2011)];  tracks from GP and GS penetrated to the North until reaching 50°N, where the Jet Stream locates, and then turn to the East. *Lu et al.* (2013) showed remarkably similar trajectories of the two major sources, i.e. GP and GS, associated with the more than 100-yr heavy precipitation induced flood event in Western France in January 1995.

Figure 2 (g) – (l) shows the TME tracks that contributed to the April 2005 flood event (Table 1). The major moisture sources identified are GP and GS. There were a few TME tracks from PE, which contributed only to the early stage of the heavy rainfall, indicated by both the number of tracks and moisture release. It took averagely 6 – 7 days, depending on the carrying wind speed and season, for

TME tracks born in PE to reach the N.E. USA. GP and GS are thus the major contributors. They are both very active in MAM (*Knippertz and Wernli*, 2010). The PE TME (Figure 2 (g) – (h)) propagated further north reaching 50°N, different from the winter event shown in Figure 2 (a) – (f).  The same is the case for the TME born in GP and GS (Figure 2 (g) – (l)). This may be associated with the beginning of the seasonal northward shift of the jet stream (*Laing and Michael Fritsch*, 1997). The moisture

released from the TME was a significant fraction of the moisture carried by the tracks, and Figure 2 (h), (j) – (l) show that the most extensive releases were occurred in the N.E. USA.

Figure 2 (m) – (r) shows the TME associated with the June 1998 flood event (Table 1). The major moisture sources identified are GP and GS, with some from the tropical East Pacific. The most notable

feature in Figure 2 (m) – (r) is the well-organized spatial trajectory of all TME tracks that bring moisture from the warmer tropical ocean to the Great Plains, starting to release moisture in the Mississippi river basin before reaching N.E. USA. And we found that a flood event in Southwest Iowa occurred before this exemplified June 1998 N.E. USA flood. The TME (Figure 2 (m) – (r)) followed the





path passing both of the flooded regions in a consistent order. Note that only TME tracks that finally entered N.E. USA have been retained in Figure 2 (m) – (r), and there were additional TME tracks that were responsible for the Southwest Iowa floods but they did not eventually reach the N.E. USA. Due to the longer distance of these trajectories and the weaker westerly in summer, it took longer for the TME

5   to reach N.E. USA after they were born. The trajectories of TME exiting the N.E. USA were different in Figure 2 (m) & (n) and Figure 2 (q) & (r). The TME in (Figure 2 (m) & (n)) went further north to Quebec, Canada; while the ones born on the 21$^{st}$ and 22$^{nd}$ (Figure 2 (q) & (r)) followed a wavelike trajectory starting from the Great Lakes. The changing trajectories were likely associated with synoptic transients (*Lu et al.*, 2013).

Figure 2 (s) – (x) shows the TME tracks that contributed to the October 2005 flood event (Table 1). The major moisture sources identified are GP and GS. The most notable feature in Figure 2 (s) – (x) is the similarity to the trajectories of those born in GS in DJF (Figure 2 (a) – (f)). TME born in GS remained in the oceanic sector before reaching the N.E. USA, which resulted in a continuous recharging of

15   moisture to the tracks. At the same time in October 2005, it was reported that remnants of Tropical Storm Tammy and Subtropical Depression Twenty-two merged with incoming continental cold fronts to produce torrential rains over N.E. USA. The trajectories of the GS TME are consistent with these storms. Such a feature is not typically consistent with ARs. The release of the moisture was concentrated in the N.E. USA area.

The four flood event examples show that in the N.E. USA, TME may be closely related to floods year-around with varying major moisture sources and trajectories. The major year-around moisture sources shown in these four events are GP and GS with some contributions from PE in DJF.

**4 TME birth and ENSO**

25   In each season, TME from different moisture sources were seen to be associated with floods in the N.E. USA. The seasonality of and interannual variations in TME birth are analyzed in this section. Figure 3



presents the seasonality of and the interannual variations in the TME born in the four major sources under different ENSO phases. All four sources show a strong seasonality of the TME birth process.

GP is active year-around and peaks in June (Figure 3 (a)). The largest divergence between El Niño and

La Niña conditions occurs in Oct – Jan, with enhanced TME under El Niño and suppressed TME under La Niña conditions. A t-test for the difference in mean GP TME counts, for those born in Oct – Jan, considering unequal variance in each phase, i.e., El Niño and La Niña phases, yields a p-value of 0.008 for the null hypothesis of no difference.

GS (Figure 3 (b)) is relatively less active than GP, and has a weaker seasonality with multiple peaks in DJF, June and October. The largest divergence between El Niño and La Niña phases occurs in Nov – Dec (p level from the t-test is 0.048), and June – July (p level from the t-test is 0.013) when TME is enhanced in the El Niño phase, and in Mar – April (p level from the t-test is 0.050) when TME is enhanced in the La Niña phase (Figure 3 (b)). It is interesting that TME is suppressed in both El Niño

and La Niña phases relative to the Neutral phase in JFM, and enhanced in Aug – Oct.

PE (Figure 3 (c)) and WP (Figure 3 (d)) have very strong seasonality, evidenced by their large variances of TME born in different months. The two have opposite peaking seasons: WP (Figure 3 (d)) is very active in summer, when it is the monsoon season for East Asia. PE (Figure 3 (c)) is active in winter,

when the 'Pineapple Express' ARs are the most active and affect the west coast of USA. There is a persistent increase in WP TME (Figure 3 (d)) in Feb – Jul (p level from the t-test is 0.09) in the La Niña phase. For PE (Figure 3 (c)) the situation is mixed, with enhancement under El Niño in January and October (p level from the t-test is 0.06), but under La Niña in December as the largest divergence.

**5 TME entrance and Atmospheric Circulation Patterns**

The origins of the TME entering N.E. USA vary seasonally, due in part to the seasonality of the TME birth, and in part to the seasonal and interannual changes in atmospheric circulation patterns. The corresponding data for the total number of tracks entering by calendar month, the total number of tracks





from the four sources considered here, and the annual total for each source are presented in Table 2. The probability P(Source|NE) that the TME that entered the NE in a given month comes from a particular source, and its interannual variation are illustrated in Figure 4.

First we note that the four sources considered account for 85% of all tracks entering the N.E. USA on an annual average basis, varying cyclically from a minimum of 70% in December to a maximum of 98% in July (Table 2). The number of TME tracks entering peaks in the winter with a secondary maximum in June. On an annual basis approximately 73% of the tracks come from GP, 14% from GS, 3% from WP and 11% from PE. Consequently, it is not a surprise that on average, 45% (Nov) to 79% (June) of the
tracks on a monthly basis come from GP. The fraction coming from GS increases from June to October, peaking in August (28%). PE's contributions are important in November to February (~20% of the tracks), and WP is a weak spring/fall contributor accounting for a maximum of 4% of the tracks in December and February. In summary, GP is important year round, but particularly in April – July, GS has increased contributions in June – October, and PE in November – February, with WP a possible
contributor in winter.

From Figure 4, we see that GP accounts for a relatively stable seasonality of tracks entering across the year, with a consistent peak in June (~80%), and minimum in Nov – Feb (~45%) and high interannual variability in August, September and November. GS is an important source in July – September with
20 high interannual variability. PE is primarily a contributor in Nov – Feb, while WP has a low contribution throughout the year with relatively high variability across years in fall and spring.
Tracks entering from GP into the N.E. USA are positively correlated at a significance level of 0.05, with those entering from GS in January (correlation=0.62), February (0.79), March (0.71), September (0.37), November (0.51) and December (0.68); with PE tracks entering in November (0.37) and December
(0.18), and with WP tracks in January (0.52) and February (0.39). Tracks entering from GS and PE are negatively correlated in January (-0.32) and October (-0.38). Tracks entering from PE and WP are positively correlated in December (0.26).





The seasonal and interannual variability in the relative contributions of the tracks from different sources may be due to changes in the TME birth or TME steering characteristics. To develop some understanding of these issues, the statistics of TME birth and the conditional probability of tracks born in a region entering the North East, P(NE|Source) are summarized in Table 3 and Figure 5.

First, note that the total tracks born aggregated across the four source regions have a pronounced seasonality with a maximum in March, June and December and a minimum in August (Table 3 & Figure 5(a)). In terms of the proportion of tracks born in the four source regions that essentially entered N.E. USA, on average only 13% of these tracks enter the N.E. USA, with a minimum of 8% in August, a maximum of 20% in November and December and an active 18% in JFM. Consequently, the seasonal cycle of the tracks entering the N.E. USA has a minimum in August, with peaks in December to March and in June.

On average, 42% of the GP TME enter N.E. USA with small variation from month to month, slightly lower in May-August, and higher in March-April and September to December (Table 3 & Figure 5(b)). The GP birth, i.e. P(GP) (Figure 3 (a)), varies across the year with a peak in June, while its entrance, i.e. P(NE|GP) (Figure 6 (a)), undergoes small changes through the year but has strong interannual variations in January and October. A t-test for the difference in mean GP TME entrance counts considering unequal variance in each phase, i.e., El Niño and La Niña phases, yields no difference. Therefore, the variations in the birth process of GP tracks over the year dominate the contributions to the seasonality of the TME that enter the N.E. USA.

GS TME entering N.E. USA accounts for 13% of the tracks born with variations from 8% in May to 23% in August (Table 3 & Figure 5(b)). GS TME entrance is active in June through September, while P(NE|GS) peaks at the same period during El Niño phase and neutral phase of ENSO (Figure 6 (b)), with anomalous decreases in July and August in La Niña phases. While the GS birth (Figure 3 (b)) has multiple peaks in DJF, June and October, the June peak is suppressed in the La Niña phase relative to the Neutral and El Niño phases. December GS birth is enhanced during El Niño phase, while January



and February are suppressed during both El Niño and La Niña years. In La Niña years, the GS birth drops in June, July (GS entrance also drops in La Niña in July (Figure 6 (b)), November with two mild peaks in October and February. Thus, it appears that ENSO may influence the birth process more than the steering process for GS TME tracks coming to the N.E. USA.

PE TME begins its entrance in October until March (Figure 5(b)) with its peak contribution in November (31%) and December (28%) (Table 3). PE birth is active at the same time period with strong interannual variations. In La Niña years, there is a decrease of P(NE|PE) in October to March, while it is also suppressed in the El Niño years in Oct – Dec (Figure 6 (c)).

WP TME entrance is the least of the four sources through the year (Figure 5(b)), though its birth peaks from May to September (Figure 3(d)). P(NE|WP) is low and the separation by ENSO episodes is minor.

In summary, it appears that ENSO's dominant influence on the interannual variations in the birth

15 process for GP and GS, and on the steering and birth process for PE. The expression of the ENSO influence varies by time of year in both birth and steering. Interannual variability in P(NE|GP) is highest in May – June, but does not appear to be related to ENSO. The variability in P(NE|GS) is highest for August – September, and again does not have a clear ENSO influence. For P(NE|PE) the Oct – March period is the most active and does seem to be influenced by ENSO.

Figure 7 and Figure 8 provide the composites of daily sea level pressure anomalies of the top 10% TME active entrance days and the top 10% TME inactive entrance days on a monthly basis to illustrate the differences of the atmospheric circulation patterns associated with the activity of TME's entrance. We consider the total number of TME that from all sources including regions that outside the four major

25 sources. The top 10% TME active days are determined by the total number of TME entering N.E. USA by finding the days that have TME exceeds the 90% percentile of the daily TME tracks for that month over the 22 years (1989 – 2010). The top 10% TME inactive entering days are determined by finding the days that have TME below the 10% percentile of the daily TME tracks for that month over the 22





years. To assist the comparison of the associated circulation patterns, we have corresponding active and inactive composites side by side (left panel: active TME entrance; right panel: inactive TME entrance) for each month (Jan – Jun in Figure 7 and Jul – Dec in Figure 8). Winter (DJF) active TME entrance is observed to be associated with low-wavenumber SLPa patterns around 60ºN; spring (MAM) active

TME are associated with lows in the Great Plain east to the Rocky Mountains; summer (JJA) TME entrance has less association with large circulation patterns; Fall (SON) TME active entrance days are associated with lows in Great Plains; its inactive entrance associated with high-wavenumber blocks in the mid-latitudes in November (Figure 8(k)).

## 6 TME and Extreme Precipitation

The moisture release from the tracks in N.E. USA is highly correlated with the total number of TME entering N.E. USA (concurrent correlation between number of TME entering and $\Delta Q$ in N.E. USA is 0.88 (p-value $< 10^{-4}$). We estimated the conditional density of daily precipitation given total daily change of specific humidity ($\Delta Q$ in Eq. (2)) of TME entering the N.E. USA using the local polynomial density estimation with the R package '*hdrcde*' (*Kim et al.*, 2011). Figure 9 shows that as the daily

moisture release ($\Delta Q$) by the TME increases, the daily precipitation increases with a shift in the conditional distribution that is marked beyond a threshold of $\Delta Q$ of about 3500 g/Kg. This observation based on data pooled over the whole year motivates a seasonal analysis of the association between extreme TME and extreme precipitation for different seasons, which is illustrated in Figure 10. The boxplot of the moisture releases ($\Delta Q$) from the TME to N.E. USA are significantly different (p-value

0.01 to 0.001) with two-sample Kolmogorov-Smirnov test) given extreme or non-extreme rainfall states for all the four seasons. An extreme rainfall event is defined as one exceeding the 99[th] percentile of daily rainfall (including days with no or trace rain) in that season, e.g., for an extreme rainfall event in DJF must exceed the 99[th] percentile of daily rainfall amounts in all the Dec, Jan and Feb over the 22 years. The seasonal 99[th] percentile thresholds are 12.7cm (DJF), 12.2cm (MAM), 11.4cm (JJA) and

15.0cm (SON). For all the four seasons, non-extreme rainfall days have their average moisture releases from TME around 0 g Kg$^{-1}$ with thin, long tails.



We further examined the ENSO influence on the extreme rainfall events occurrence. The number of extreme events fall into each ENSO episode is tabulated in Table 4. Across the whole year, there are 36 extreme events in La Niña years, with 16 events in El Niño years out of 83 total events at the 99[th] of daily rainfall by season. Extremes in June have 5 out of total 7 in La Niña years, against 0 in El Niño

5    years. March, April and September also have similar observations. However, due to the limitation of the sample size, the statistical significance is weak (e.g., p=0.18 for June, even though we have 0 El Niño and 5 La Niña cases).

On a monthly basis, the numbers of TME entering the N.E. USA for extreme precipitation events is

consistently larger than those for non-extreme events. The definition of monthly extreme rainfall events is the same as that for seasonal extreme events except that the 99[th] percentile thresholds are taken for each month. Table 5 provides the ratio of the average TME tracks entering N.E. USA from each source for each month. The ratio is the average over all extreme events divided by the average over all non-extreme events. As the year-around major source, GP shows consistent intensification of TME entrance

on extreme rainy days. The average TME from GP on extreme rainy days are 4 to 7 times of the average on non-extreme days. The difference of TME counts for days in the above and below 99[th] percentile rainfall categories is statistically significant with a p-value of less than 0.001 for the null hypothesis of no difference. The second major source, GS shows year-around intensification of TME entrance except for July when the ratio is close to 1. The intensification is stronger in GS than GP due to the fact that the

average TME from GS on non-extreme rainfall days are less than those from GP, but their average TME on extreme rainy days are comparable. This suggests that for extreme rainfall events, GP and GS are both important. PE and WP are both active from Oct to Apr (Figure 6 (c) & (d)) and contribute very little from May to Sep. The average TME entrance from PE and WP are less than 3 for non-extreme rainy days from May to Sep/Aug, and hence the corresponding entries in Table 5 are left blank.   The

average number of TME from PE entering N.E. USA in active months ranges from 10 to 34; and it ranges from 3 to 6 from WP. Although the ratios for PE in Mar and for WP are large, their contributions to extreme rainfall events are less than those from GP and GS.





## 7 Summary and Discussion

The key findings of the paper are summarized as follows:

1. The N.E. USA floods in the four seasons are closely related to TME as evidenced by the historical flood events.

2. The four major moisture sources of TME account for approximately 85% of all the TME entering the N.E. USA. The birth processes of the four are relatively independent, except for moderate association between GP and GS in some months. They all have strong seasonality and interannual variation, which determine their contributions to N.E. USA. Their overall contributions can be ordered as GP>GS>PE>WP, with GP and GS as the year-around sources,

and PE active in winter, and WP the smallest contributor.

3. Depending on the month, some of the interannual variations of TME birth are associated with ENSO phases. Since GP is the dominant contributor and year-around source, the influence of ENSO on GP TME birth affects the TME entrance to N.E. USA. The intensification of TME born from October to January during the ENSO warm phase suggests that more TME could

potentially enter N.E. USA, which may result in more moisture release and precipitation. Since the ENSO warm phase also leads to an intensification of the GS birth process, the two major sources would contribute more TME potentially, if the steering mechanism is not changed, to bring the tracks to the N.E. USA. The GP and GS TME track entrance is highly correlated over several months, and this may reflect the common influences on the birth and steering process.

4. The seasonal and interannual variations in atmospheric circulation patterns also play an important role in determining the TME's entrance to N.E. USA. Aggregating cross the four major sources, the annual maximum of TME entrance occurs in June and the minimum occurs in winter. The order of importance of the four sources is consistent with observations in the floods examples in the beginning: GP>GS>PE>WP. However different from the ENSO effect on the

birth process, no significant difference was observed among different ENSO episodes for the entrance given a birth source, suggesting the ENSO has more influence on birth than steering mechanism. The composite SLPa of active TME days and inactive TME days suggests that low wave number patterns of atmospheric circulation mark the anomalous transport. Depending on



the calendar month, these patterns are not always symmetric for active vs inactive TME periods suggesting that the circulation processes driving the TME to the N.E. USA are nonlinear.

5. The number of tracks entering and the associated moisture release are highly correlated. This translates into a strong influence of active TME periods on the occurrence of extreme rainfall. The distribution of TME for extreme rainfall events (>the 99[th] percentile of daily rainfall) is significantly different from the one given non-extreme events, suggesting the important role of TME in determining extreme precipitation. This argument carries forward into shifts in extreme rainfall event occurrence across different ENSO phases as they influence the TME birth and steering.

The study of atmospheric rivers as an influence on floods induced by extreme precipitation has significantly enhanced the interaction between hydrologists and climate scientists towards an improved understanding of the synoptic and climatological factors that govern such phenomena. The N.E. USA has not been the target of many of these investigations, partly because of the way atmospheric rivers have been defined. The Knippertz and Wernli work established the broader role of tropical moisture exports in the climatology of mid-latitude precipitation, and provided a data base that allows a consistent exploration of the Lagrangian transport of moisture from the tropics to the mid-latitudes. Our initial work on a flood over Western France in 1995 revealed that at synoptic scales systematic organization of moisture as reflected in the data not including 1995 could be identified by a simple statistical model and used effectively for an out of sample prediction of the extreme precipitation in January 1995 that led to the flood event. Several of the moisture tracks associated with that flood event also had a moisture release over the N.E. USA, stimulating the work reported in this paper. Here, we explore climatological aspects of the links between TME birth, steering, moisture release and extreme precipitation, providing the first such chronology of the year round links between these factors, as well as the potential links to ENSO.

Given the short record of TME available to us, a structured exploration of interannual variability and an association with ENSO and lower frequency phenomenon was not possible. However, we expect to



extend the work reported here in multiple directions. First, we expect that a national analysis will be much more informative as to the spatial and temporal shifts in TME influence on extreme precipitation by season and as driven by identified low frequency climate mechanisms. Developing a formal spatio-temporal model that considers the connectivity across such a source-destination network and its

modulation by atmospheric and ocean circulation precursors is a challenge worthy of pursuit. As we develop such a model, we expect to utilize longer atmospheric-reanalysis data and to develop a longer TME record from it using LAGRANTO as was done by Knippertz and Wernli using the ERA-Interim data. Such a development may provide a much better empirical understanding of extreme precipitation dynamics across the USA, and thus provide an important diagnostic tool for the performance of climate

models for seasonal forecasting or for climate change simulations.

**Acknowledgements**: The floods record is available at the website of the Dartmouth Flood Observatory (DFO: http://floodobservatory.colorado.edu/ ). The TME data is provided by P. Knippertz, documented in *Knippertz and Wernli* (2010) and *Knippertz et al.* (2013). The NCEP/NCAR Reanalysis dataset is

provided by the NOAA/OAR/ESRL PSD, Boulder, Colorado, USA, from their Web site at http://www.esrl.noaa.gov/psd/ . The Oceanic Niño Index (ONI) is provided by NOAA/National Weather Service, NOAA Center for Weather and Climate Prediction, Climate Prediction Center, from their website at http://www.cpc.ncep.noaa.gov/.





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





**Tables**

**Table 1**:Facts of the example flood events in the each of the four seasons, record are from Dartmouth Flood Observatory Archive.

| Season | DJF | MAM | JJA | SON |
|---|---|---|---|---|
| Location | US: New York, Pennsylvania, Ohio, West Virginia, New Jersey; Canada: Chateauguay, Quebec | New York, New Jersey, Delaware River | Green Mountain to Bradford, Vermont and New Hampshire, Mad River, White River and Ayers Brook | Southwest New Hampshire, Vermont, New Jersey, Connecticut |
| Time | 1996, January | 2005, April | 1998, June | 2005, October |
| Facts of the Floods | 2 successive flood events occurred in N.E. USA and Quebec Canada, triggered by heavy rain and rapid snowmelt with anomalous warmer temperature | 7 inches rainfall total with antecedent wet condition from Tropical Storm 'Ivan', worst flood event in the 50 years | 8 inches rainfall total, flows exceeded the 100-year flood | 12 inches of rainfall total, intensive rainfall of 7 inches within 30 hours period, flows exceeded the estimated 500-yr flood |
| Moisture Sources | GP, GS and PE | GP, GS, very few PE | GP, GS, few from East Pacific | GP and GS |



**Table 2:** Conditional probability of TME sources given the tracks entering N.E. USA region [39°N – 48°N, 66°W-82°W] in each month through a year, the conditional probability is averaged over 22 years of data, from 1989 to 2010. The last two columns correspond to number of tracks from the four major sources and number of tracks from all sources including the four and anywhere outside these.

| Month | \multicolumn{5}{c}{Conditional Probability of TME sources given Tracks Entered NE: **P(Source\|NE)**} | |
|---|---|---|---|---|---|---|
| Month | P(GP\|NE) | P(GS\|NE) | P(PE\|NE) | P(WP\|NE) | #Tracks (PE,WP,GS,GP) | #Tracks |
| Jan | 0.484 | 0.072 | 0.211 | 0.027 | 3590 | 4622 |
| Feb | 0.513 | 0.083 | 0.161 | 0.041 | 3208 | 3933 |
| Mar | 0.655 | 0.071 | 0.107 | 0.031 | 3758 | 4358 |
| Apr | 0.745 | 0.063 | 0.047 | 0.028 | 3060 | 3488 |
| May | 0.768 | 0.091 | 0.040 | 0.023 | 2887 | 3118 |
| Jun | 0.787 | 0.132 | 0.003 | 0.017 | 3757 | 3956 |
| Jul | 0.781 | 0.202 | 0.000 | 0.003 | 2631 | 2682 |
| Aug | 0.677 | 0.277 | 0.001 | 0.019 | 2114 | 2171 |
| Sep | 0.599 | 0.242 | 0.020 | 0.061 | 2764 | 2969 |
| Oct | 0.556 | 0.132 | 0.083 | 0.032 | 2485 | 3004 |
| Nov | 0.454 | 0.068 | 0.173 | 0.034 | 2859 | 3861 |
| Dec | 0.461 | 0.068 | 0.180 | 0.042 | 3582 | 5095 |
| Annual Total #Tracks | 26670 | 4974 | 3920 | 1130 | 36694 | 43256 |



**Table 3:** Conditional probability of tracks entering the N.E. USA [39°N – 48°N, 66°W-82°W] region, given tracks born in the four major sources in each month through a year, the conditional probability is averaged over 22 years of data, from 1989 to 2010.

| | Conditional Probability of Tracks entered NE given born in the sources:  P(NE\|Source) | | | | | |
|---|---|---|---|---|---|---|
| **Month** | **P(NE\|GP)** | **P(NE\|GS)** | **P(NE\|PE)** | **P(NE\|WP)** | **#Tracks Entered NE** | **#Tracks (PE,WP,GS,GP)** |
| **Jan** | 0.392 | 0.119 | 0.178 | 0.037 | 3590 | 18906 |
| **Feb** | 0.349 | 0.098 | 0.117 | 0.036 | 3208 | 18510 |
| **Mar** | 0.469 | 0.118 | 0.085 | 0.015 | 3758 | 21237 |
| **Apr** | 0.496 | 0.085 | 0.033 | 0.011 | 3060 | 21526 |
| **May** | 0.368 | 0.080 | 0.032 | 0.006 | 2887 | 28879 |
| **Jun** | 0.382 | 0.131 | 0.001 | 0.004 | 3757 | 36917 |
| **Jul** | 0.339 | 0.154 | 0.000 | 0.000 | 2631 | 32381 |
| **Aug** | 0.373 | 0.225 | 0.001 | 0.002 | 2114 | 24895 |
| **Sep** | 0.465 | 0.213 | 0.016 | 0.009 | 2764 | 23217 |
| **Oct** | 0.470 | 0.094 | 0.084 | 0.011 | 2485 | 19714 |
| **Nov** | 0.513 | 0.109 | 0.313 | 0.022 | 2859 | 14215 |
| **Dec** | 0.422 | 0.116 | 0.282 | 0.044 | 3582 | 17569 |
| **Annual Total #Tracks** | 26670 | 4974 | 3920 | 1130 | 36694 | 277965 |




**Table 4:** ENSO influence on the number extreme rainfall events, i.e. exceeding the 99[th] percentile threshold of that calendar month over the 22 years data period

| Month | 99[th] Rainfall (cm) | El Niño | La Niña | Neutral |
|:---:|:---:|:---:|:---:|:---:|
| Jan | 14.7 | 1 | 3 | 3 |
| Feb | 11.3 | 3 | 1 | 2 |
| Mar | 12.8 | 1 | 5 | 1 |
| Apr | 12.9 | 1 | 4 | 2 |
| May | 11.1 | 2 | 3 | 2 |
| Jun | 11.3 | 0 | 5 | 2 |
| Jul | 11.7 | 2 | 1 | 4 |
| Aug | 11.0 | 1 | 2 | 4 |
| Sep | 14.5 | 1 | 4 | 2 |
| Oct | 15.0 | 3 | 2 | 2 |
| Nov | 16.0 | 0 | 3 | 4 |
| Dec | 13.0 | 1 | 3 | 3 |
| No. of Years | | 5 | 8 | 9 |



**Table 5:** The ratio of average TME tracks entering N.E. USA from each source: extreme rainfall events over non-extreme rainfall events; extreme rainfall events are defined as the ones exceeding the 99[th] percentile of daily rainfall for each month; some entries are blank because the average number of tracks are less than 3 for non-extreme events; the differences between the two samples (i.e. Extreme rainfall case vs. Non-extreme rainfall case) are statistically significant (P-value < 0.001 with t-test) for the white entries; insignificant months are marked grey.

| Month | GP | GS | PE | WP |
|---|---|---|---|---|
| Jan | 4.08 | 9.39 | 2.80 | 6.46 |
| Feb | 6.40 | 7.08 | 8.60 | 8.92 |
| Mar | 4.02 | 7.72 | 10.67 | 9.85 |
| Apr | 4.00 | 8.33 | 2.94 | 25.67 |
| May | 3.30 | 8.91 | | |
| Jun | 4.19 | 5.69 | | |
| Jul | 3.76 | 0.89 | | |
| Aug | 6.48 | 5.16 | | |
| Sep | 6.20 | 8.18 | | 2.75 |
| Oct | 4.51 | 5.88 | 5.69 | 12.78 |
| Nov | 3.50 | 4.52 | 7.06 | 9.54 |
| Dec | 4.92 | 8.87 | 4.91 | 11.74 |





**Figures**

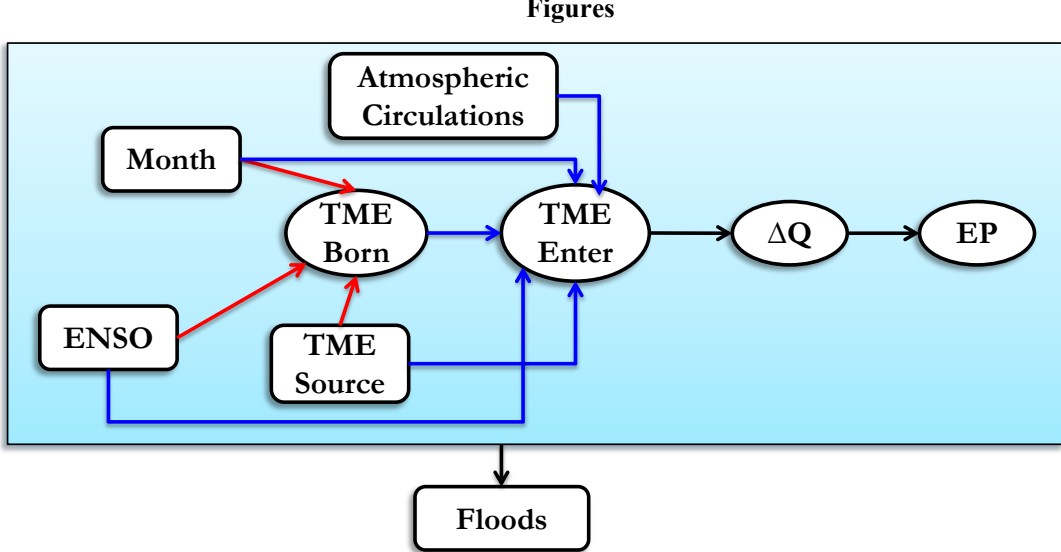

**Figure 1:** Influence Diagram of the factors considered and their proposed relationship (as the arrows directed) investigated in this paper



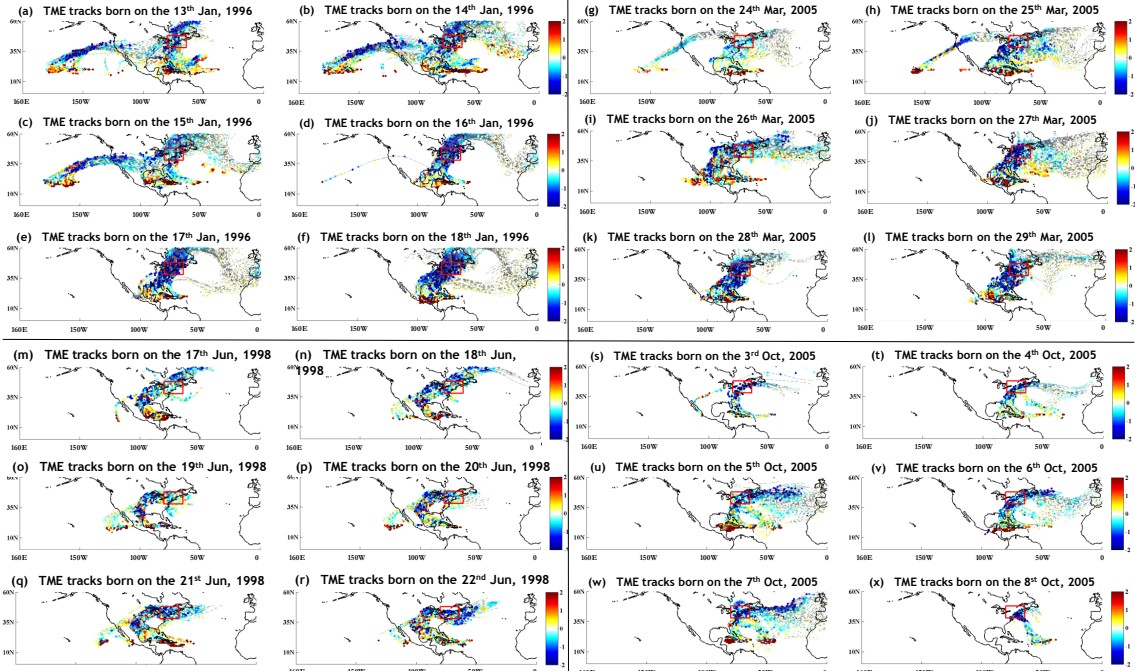

**Figure 2:** (a) – (f): DJF Flood: TME tracks that born between the 13th and the 18th Jan in 1996, two successive flood events occurred, one started on the 15th in New York, Pennsylvania, Ohio, West Virginia, New Jersey; the other started 4 days later in Chateauguay, Quebec. (g) – (l): MAM flood: TME tracks that born between the 24th and the 29th Mar in 2005; broad area including New York, New Jersey, Delaware had flood starting on the 1st of April. (m) – (r): JJA flood: TME tracks that born between the 17th and the 22nd Jul in 1998; heavy rainfall induced flood event in Green Mountain to Bradford, Vermont and New Hampshire. (s) – (x) SON flood: TME tracks that born between the 3rd and 8th Oct in 2005 associated with 10 days flooding from the 8th to the 17th of Oct in Southwest New Hampshire, Vermont, New Jersey and Connecticut. The red box highlights the N.E. USA with color indicating the changes of moisture from the tracks: red = pickup, blue= release; the red box highlights the N.E. USA; each dot corresponds to a 6-hour update of the air parcel position, i.e., tracks trajectory. The unit of the color bar is g Kg$^{-1}$. (Dartmouth Flood Observatory Archive http://floodobservatory.colorado.edu/)



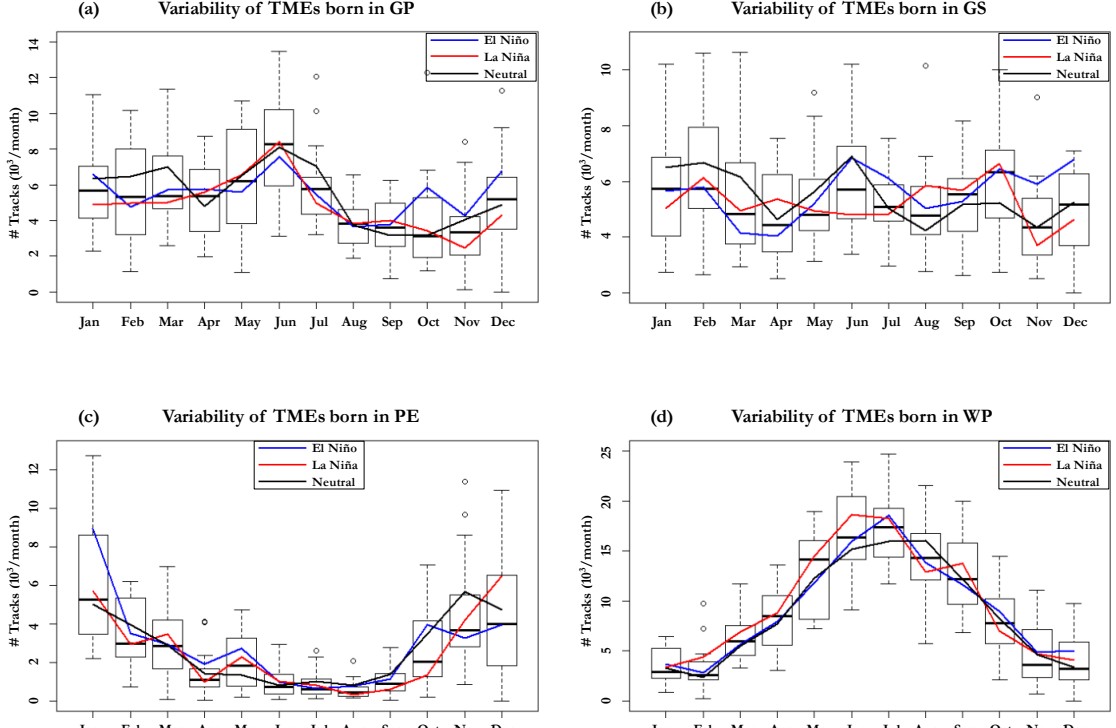

**Figure 3:** Seasonality and interannual variations (boxplot) of the monthly total number of TME born in the four major source regions (a) GP, (b) GS, (c) PE and (d) WP and the influence of ENSO scenarios on their monthly average amounts: blue is the composite of the El Niño years; red is the composite of the La Niña years; black is the composite of the neutral years.





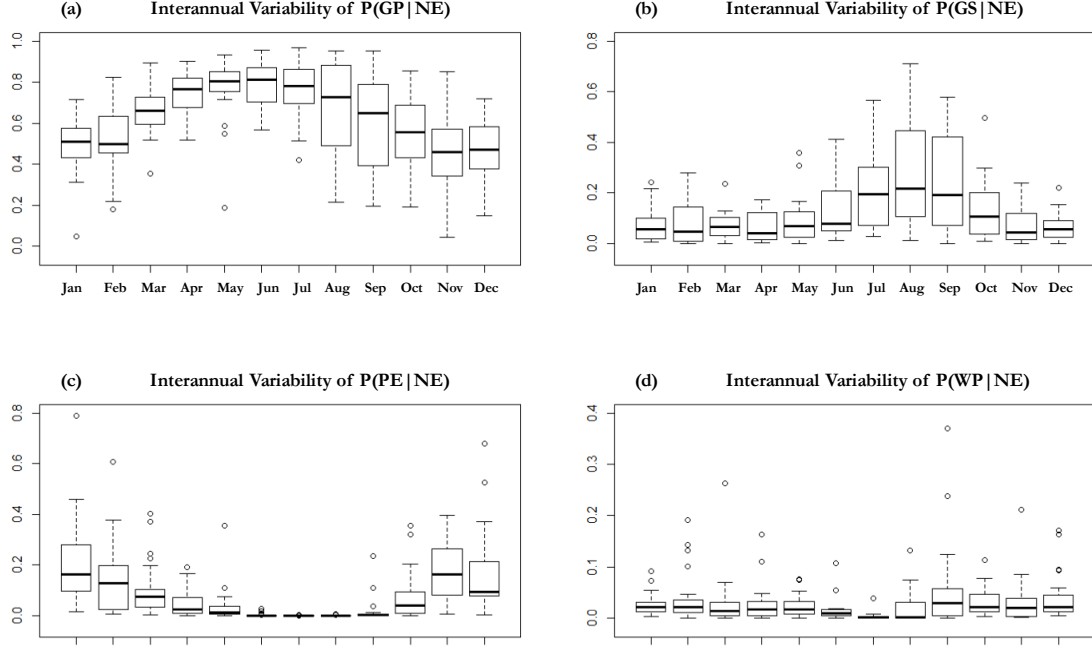

**Figure 4:** Boxplots of the conditional probabilities of TME birthplace (sources regions), given TME tracks entered N.E. USA: the interannual variability of monthly contributions of TME tracks from (a) GP, (b) GS, (c) PE and (d) WP.



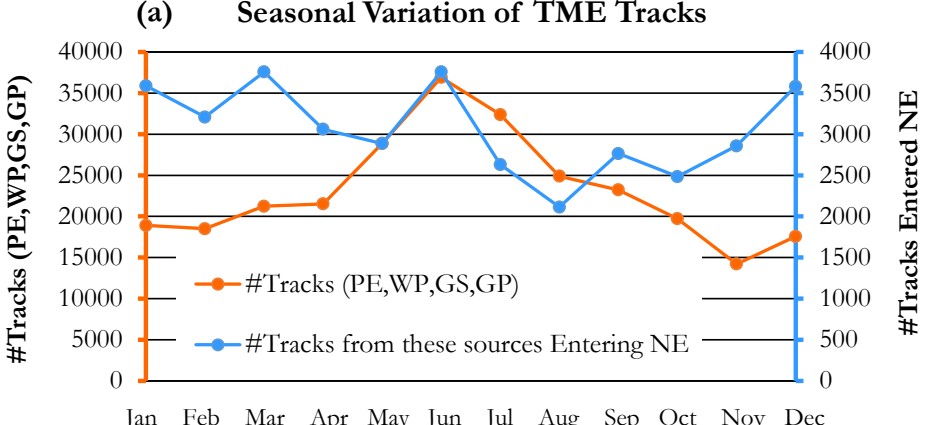

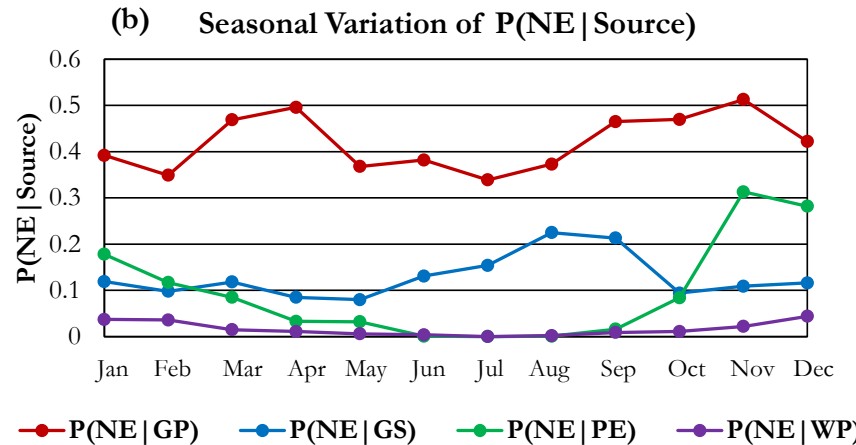

**Figure 5:** (a) Seasonal variation of the TME born in the four major sources and the seasonally varying portions of these TME entering N.E. USA, note that the TME born refer to the y-axis on the left and entrance refer to the y-axis on the right, whose scale is 1/10 of the left; (b) seasonal variation of the probability of TME entering N.E. USA given TME born in four major source regions.





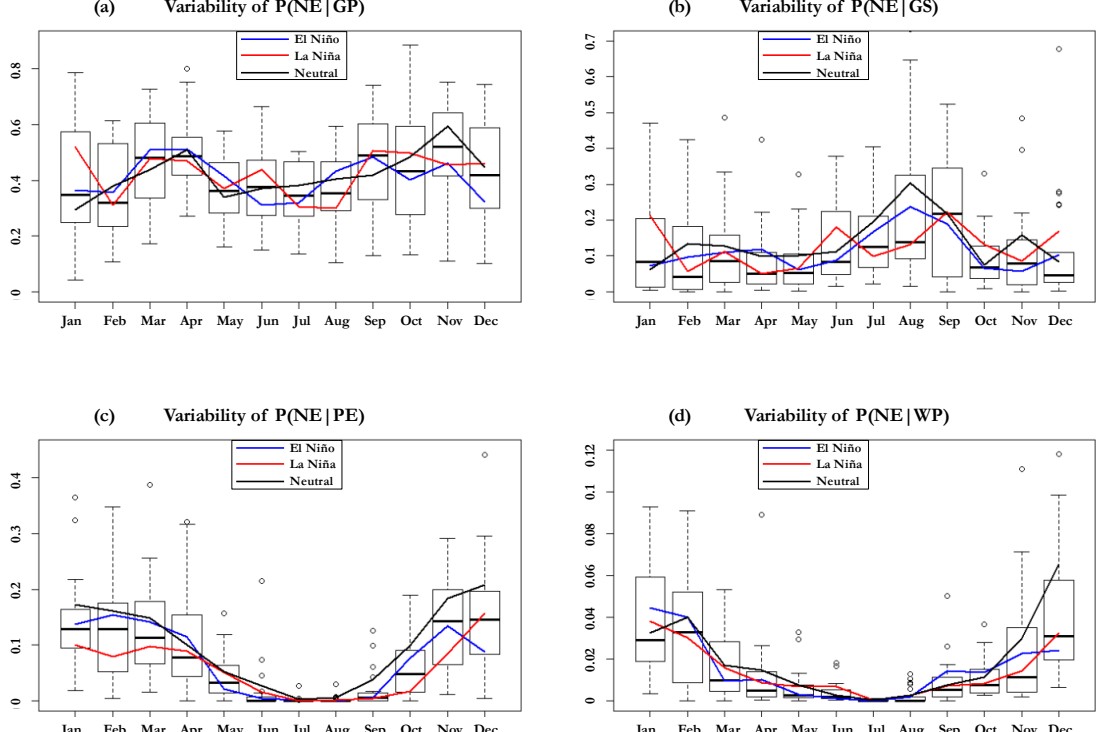

**Figure 6:** Seasonality and interannual variations of the conditional probabilities of TME Entrance to N.E. USA for each month, given TME born in the four major source regions in the tropics: (a) GP, (b) GS, (c) PE and (d) WP; and the influence of ENSO episodes: blue is the composite of the El Niño years; red is the composite of the La Niña years; black is the composite of the neutral years.





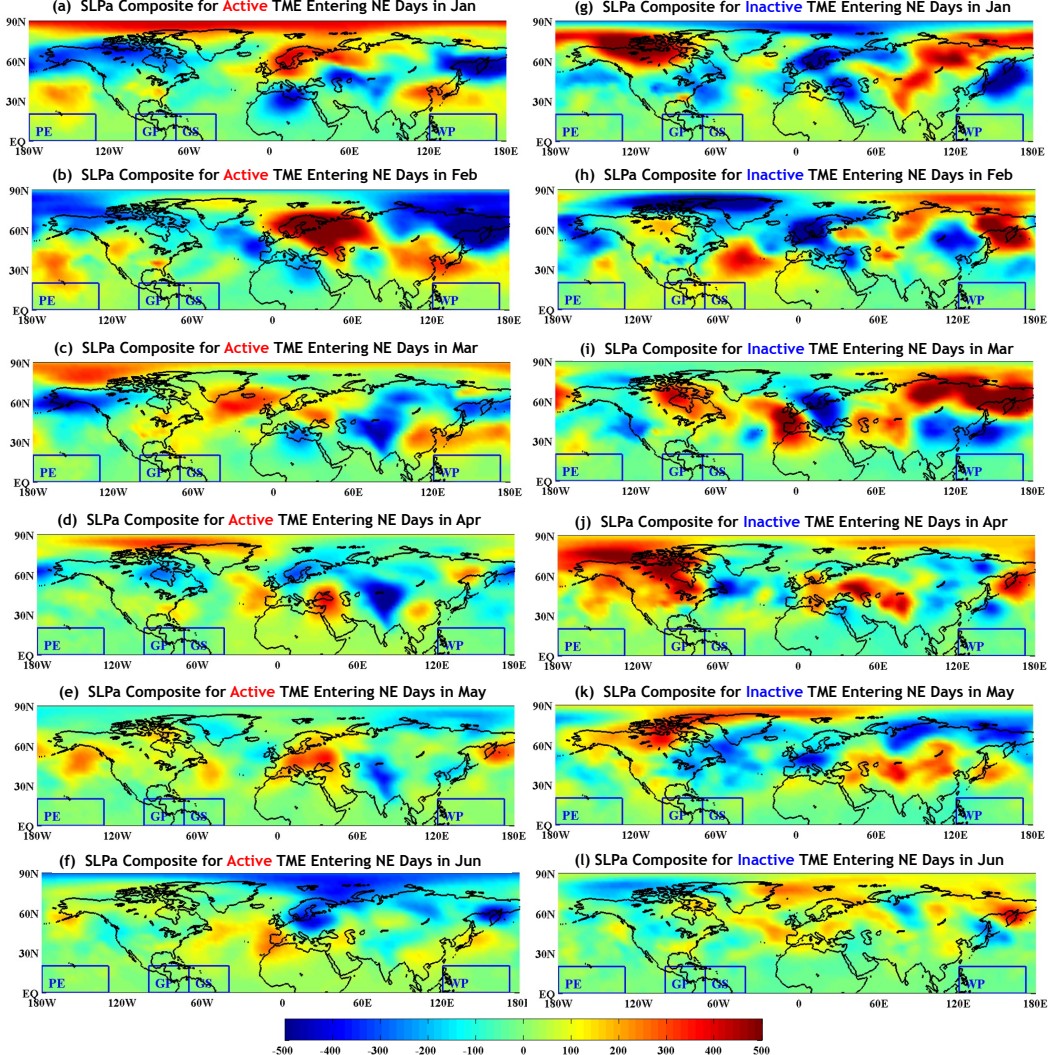

**Figure 7:** Composite of daily Sea Level Pressure anomalies of the top 10% TME active entering N.E. USA days in each month from January to June ((a) – (f)), over the 22 years (1989 -2010); and composite of daily Sea Level Pressure anomalies of the top 10% TME inactive days in each month from January to June, (g) – (l), over the 22 years (1989 -2010). TME active entering days are categorized as



the days which have the number of total TME tracks entering N.E. USA exceeding its monthly 90% percentile; the composite is done for each month, with 90% percentiles calculated for each month over the 22 years; TME Inactive entering days are categorized as the days which have the number of total TME tracks entering N.E. USA below its monthly 10% percentile; the composite is done for each month, with 10% percentiles calculated for each month over the 22 years; the four major TME sources in the tropics are marked in blue rectangular boxes with their names: PE (Pineapple Express), GP(Great Plains), GS(Gulf Stream) and WP (West Pacific).


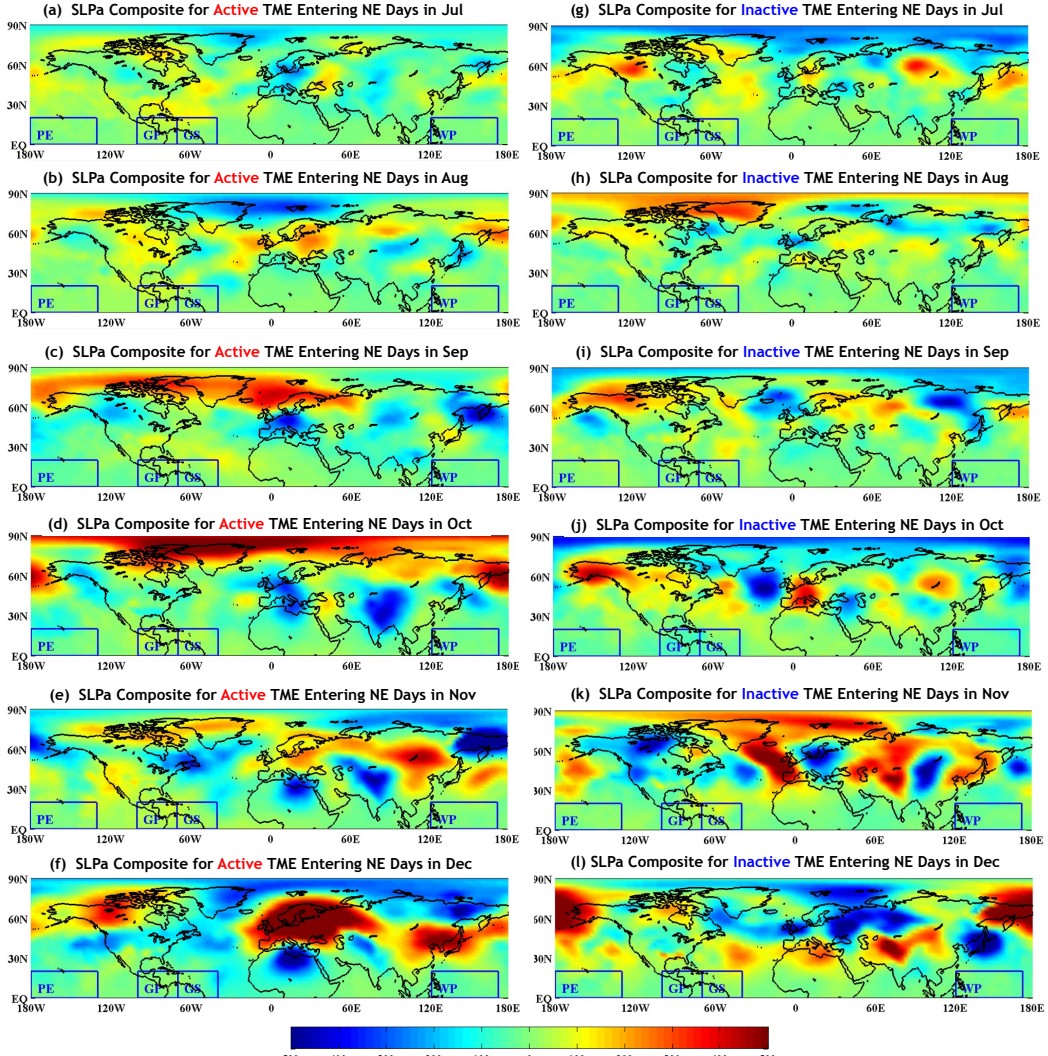

**Figure 8:** Composite of daily Sea Level Pressure anomalies of the top 10% TME active entering N.E. USA days in each month from July to December ((a) – (f)), over the 22 years (1989 -2010); and composite of daily Sea Level Pressure anomalies of the top 10% TME inactive days in each month from July to December, (g) – (l), over the 22 years (1989 -2010). TME active entering days are categorized as the days which have the number of total TME tracks entering N.E. USA exceeding its monthly 90%



percentile; the composite is done for each month, with 90% percentiles calculated for each month over the 22 years; TME Inactive entering days are categorized as the days which have the number of total TME tracks entering N.E. USA below its monthly 10% percentile; the composite is done for each month, with 10% percentiles calculated for each month over the 22 years; the four major TME sources

5   in the tropics are marked in blue rectangular boxes with their names: PE (Pineapple Express), GP(Great Plains), GS(Gulf Stream) and WP (West Pacific).





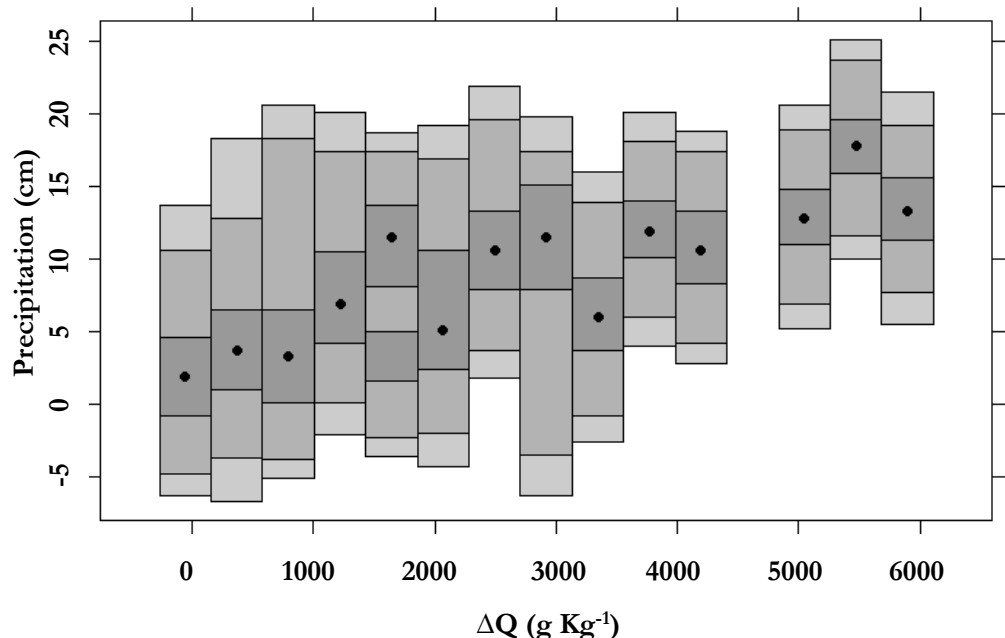

**Figure 9:** Kernel conditional density of daily precipitation given total changes of specific humidity of TME entered N.E. USA from 1989 – 2010, estimated using local polynomial. ΔQ is calculated by Eq. (1).




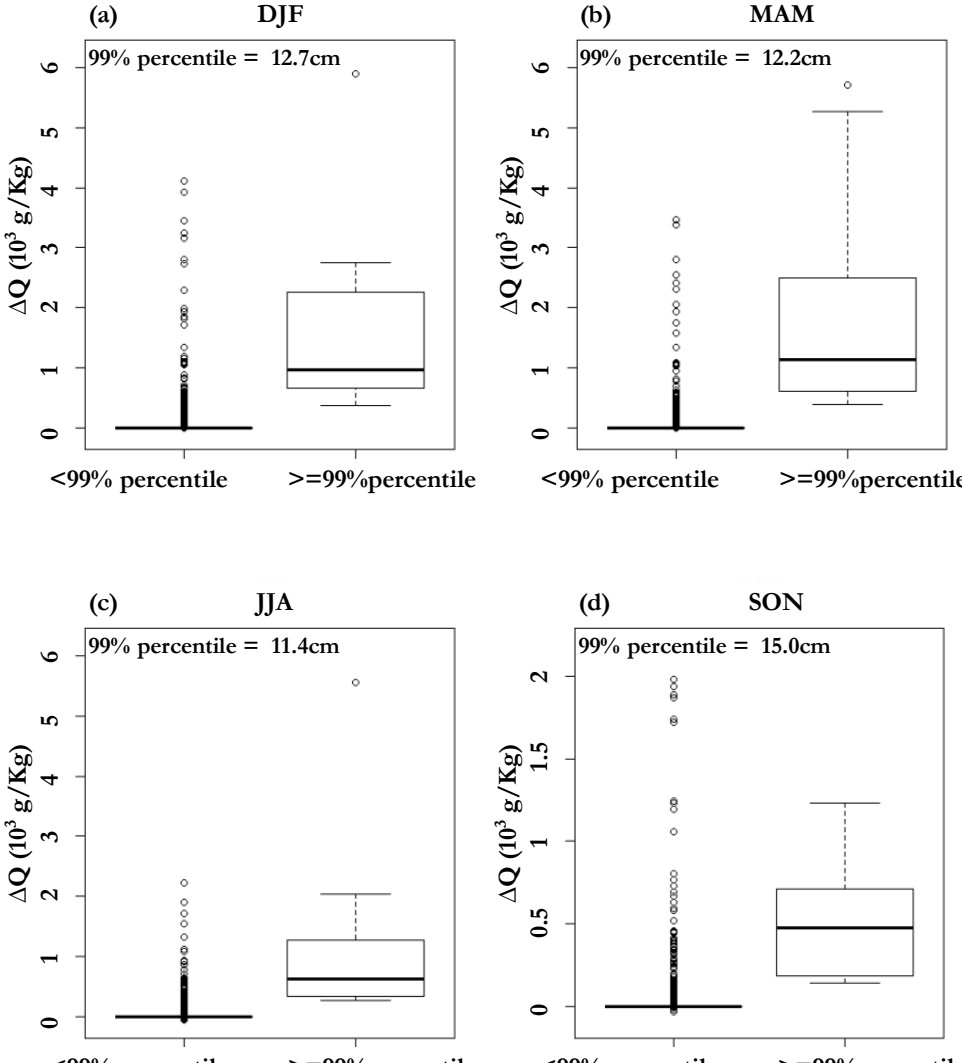

**Figure 10:** Boxplots of the moisture releases ($\Delta Q$, unit: g Kg$^{-1}$) from the TME to N.E. USA for each season given concurrent extreme (exceeding seasonal 99[th] percentile threshold, in blue) or non-extreme





rainfall states, in red. The seasonal 99th percentile thresholds are (a) DJF, 12.7cm; (b) MAM, 12.2cm; (c) JJA, 11.4cm and (d) SON, 15.0cm.

**Figure Captions**

Figure 1: Influence Diagram of the factors considered and their proposed relationship (as the arrows directed) investigated in this paper

Figure 2: (a) – (f): DJF Flood: TME tracks that born between the 13th and the 18th Jan in 1996, two successive flood events occurred, one started on the 15th in New York, Pennsylvania, Ohio, West Virginia, New Jersey; the other started 4 days later in Chateauguay, Quebec. (g) – (l): MAM flood: TME tracks that born between the 24th and the 29th Mar in 2005; broad area including New York, New Jersey, Delaware had flood starting on the 1st of April. (m) – (r): JJA flood: TME tracks that born between the 17th and the 22nd Jul in 1998; heavy rainfall induced flood event in Green Mountain to Bradford, Vermont and New Hampshire. (s) – (x) SON flood: TME tracks that born between the 3rd and 8th Oct in 2005 associated with 10 days flooding from the 8th to the 17th of Oct in Southwest New Hampshire, Vermont, New Jersey and Connecticut. The red box highlights the N.E. USA with color indicating the changes of moisture from the tracks: red = pickup, blue= release; the red box highlights the N.E. USA; each dot corresponds to a 6-hour update of the air parcel position, i.e., tracks trajectory. The unit of the color bar is g Kg$^{-1}$. (Dartmouth Flood Observatory Archive http://floodobservatory.colorado.edu/)

Figure 3: Seasonality and interannual variations (boxplot) of the monthly total number of TME born in the four major source regions (a) GP, (b) GS, (c) PE and (d) WP and the influence of ENSO scenarios on their monthly average amounts: blue is the composite of the El Niño years; red is the composite of the La Niña years; black is the composite of the neutral years.

Figure 4: Boxplots of the conditional probabilities of TME birthplace (sources regions), given TME tracks entered N.E. USA: the interannual variability of monthly contributions of TME tracks from (a) GP, (b) GS, (c) PE and (d) WP.

Figure 5: (a) Seasonal variation of the TME born in the four major sources and the seasonally varying portions of these TME entering N.E. USA, note that the TME born refer to the y-axis on the left and entrance refer to the y-axis on the right, whose scale is 1/10 of the left; (b) seasonal variation of the probability of TME entering N.E. USA given TME born in four major source regions.

Figure 6: Seasonality and interannual variations of the conditional probabilities of TME Entrance to N.E. USA for each month, given TME born in the four major source regions in the tropics: (a) GP, (b) GS, (c) PE and (d) WP; and the influence of ENSO episodes: blue is the composite of the El Niño years; red is the composite of the La Niña years; black is the composite of the neutral years.



Figure 7: Composite of daily Sea Level Pressure anomalies of the top 10% TME active entering N.E. USA days in each month from January to June ((a) – (f)), over the 22 years (1989 -2010); and composite of daily Sea Level Pressure anomalies of the top 10% TME inactive days in each month from January to June, (g) – (l), over the 22 years (1989 -2010). TME active entering days are categorized as the days which have the number of total TME tracks entering N.E. USA exceeding its monthly 90% percentile; the composite is done for each month, with 90% percentiles calculated for each month over the 22 years; TME Inactive entering days are categorized as the days which have the number of total TME tracks entering N.E. USA below its monthly 10% percentile; the composite is done for each month, with 10% percentiles calculated for each month over the 22 years; the four major TME sources in the tropics are marked in blue rectangular boxes with their names: PE (Pineapple Express), GP(Great Plains), GS(Gulf Stream) and WP (West Pacific).

Figure 8: Composite of daily Sea Level Pressure anomalies of the top 10% TME active entering N.E. USA days in each month from July to December ((a) – (f)), over the 22 years (1989 -2010); and composite of daily Sea Level Pressure anomalies of the top 10% TME inactive days in each month from July to December, (g) – (l), over the 22 years (1989 -2010). TME active entering days are categorized as the days which have the number of total TME tracks entering N.E. USA exceeding its monthly 90% percentile; the composite is done for each month, with 90% percentiles calculated for each month over the 22 years; TME Inactive entering days are categorized as the days which have the number of total TME tracks entering N.E. USA below its monthly 10% percentile; the composite is done for each month, with 10% percentiles calculated for each month over the 22 years; the four major TME sources in the tropics are marked in blue rectangular boxes with their names: PE (Pineapple Express), GP(Great Plains), GS(Gulf Stream) and WP (West Pacific).

Figure 9: Kernel conditional density of daily precipitation given total changes of specific humidity of TME entered N.E. USA from 1989 – 2010, estimated using local polynomial. ΔQ is calculated by Eq. (1).

Figure 10: Boxplots of the moisture releases (ΔQ, unit: g Kg$^{-1}$) from the TME to N.E. USA for each season given concurrent extreme (exceeding seasonal 99$^{th}$ percentile threshold, in blue) or non-extreme rainfall states, in red. The seasonal 99$^{th}$ percentile thresholds are (a) DJF, 12.7cm; (b) MAM, 12.2cm; (c) JJA, 11.4cm and (d) SON, 15.0cm.