# Peer review of "Tropical Moisture Exports, Extreme Precipitation and Floods in Northeast US"

_Hydrology and Earth System Sciences, 2016_

## Referee Comment (RC1) · Anonymous Referee #1 · 28 Sep 2016

**REVIEW OF LU AND LALL**

**GENERAL COMMENTS**

This paper discusses the role of tropical moisture exports in extreme precipitation and flooding events in the Northeastern United States. The authors seem to have done a thorough investigation, but I found it difficult to assess the value of their results. My main points are:

- Lack of comprehensiveness. I found it difficult to pick up the main messages from the paper, because it is lengthy with many figures and many abbreviations. Moreover, the text is often literally saying what is already seen in the figures, but not saying what we learn from these figures. The tables are often full of data, but could often be moved to a Supplement. The methods and definitions should be given more clearly. The conclusions are not 1-to-1 related to the research questions. I'd advice the authors to seriously condense their manuscript.
- Context. What is the reason the Norteastern United States were picked as a study region? Is this part of a bigger project or something else? Why is ENSO given so much attention as previous studies showed little influence of ENSO in this region?

These main point will also come back in the specific comments below.

**SPECIFIC COMMENTS**

1 and 1.1. The section 1.1 appears to be a mixture of literature review, objectives and method description. Please follow a more common outline. Part of it should go to the introduction and another part should be in Methods and merged with 2 Data, which is a misleading name as well, because it also discusses the modelling framework.

Why is not the latest version of the LAGRANTO model used (Sprenger and Wernli, 2015)? Is backward tracking from the NE-US applied or forward tracking and then integrated? Is the LAGRANTO model run at all by the authors or did they post-process a dataset from previous work? This is all not clear to me.

P2:L10-L11: There are many more regions in the world besides the NE-US that not have been studied much in this context. Is there any specific reason why the NE-US has been chosen as a region of study?

P2:L14-24: Some suggested definitions are discussed for ARs, "moist conveyor belt" and TME, but what is the exact definition that is used in this paper?

P3:L23-25: "Each trajectory represents $3 * 10^{12}$ kg of atmospheric mass" I do not understand the (relevance of) this statement. How many parcels are released from the vertical? Why is the unit not kg m$^{-2}$?

P4:L14: I suppose that little HESS readers will be familiar with the mei-yu-baiu front. Some explanation is required for readability.

The four regions should also be outlined in a Figure.

Why isn't sea level pressure data used from the same source as the input to LAGRANTO (ERA-Interim)?

Sea level pressure and Oceanic Niño Index is being used how and why exactly?

Four paragraphs are used to discuss Fig. 2. Could this not be summarized in one table?

P9:L25-26: Is this conclusion drawn from Fig. 2? Or from somewhere else?

P10: I'd expect a general conclusion from Fig. 3, besides the discussion of the individual panels only. The influence of ENSO seems rather small in general, and definitely not statistically significant, as the neutral years or often not in between El Niño and La Niña, am I right?

7 Summary and Discussion: There appear to be 5 key findings, but it would help the reader if the 4 research questions from Page 4 are exactly answered. Moreover, this section could be named Conclusions and significantly condensed.

Figure 1: What is the meaning of the different colors of the arrows? What does EP mean? None of it is explained in the caption.

Figure 2: The panels are really tiny and difficult to read. I'd advice to split this into at least two figures or maintain only the most important panels. Please also provide this as a movie, pdf or ppt in the Supplement with each individual panel in one view.

Figure 2: What is the exact definition of a storm track being born?

Figure 2: I suppose the dots are connected to become actual tracks? This is not distinguishable from the tiny panels. Does all tracks "end" in the NE-US? If so, what is the exact definition of the "end" of a storm track?

Figure 3: I suppose that the number of tracks is not really a physical quantity, but dependent on the resolution used in LAGRANTO or am I wrong? This should be explained.

Figure 3: Do all these TMEs also go to the NE-US? If not, what is the value of this figure?

Figure 4: Is a birthplace (source region) the same as an evaporative source (e.g. Keys et al., 2012) or something else?

Figure 5b: This is very much scale-dependent and that should be mentioned.

Figure 6: P stands for what? Precipitation, probability? I suppose probability, but it is not defined. As the ENSO signal does not do much, could this whole figure not be merged with Fig. 5b to show the variability?

Results could be compared to http://cola.gmu.edu/wcr/ (Dirmeyer et al., 2009) and then looking at the St. Lawrence river basin.

I am missing the entire point of Figures 7 and 8. What do they explain?

Figure 9 and Page 14 (and other places as well): Precipitation is a flux is should be defined per unit of time. The time integrator of ΔQ is not given. See http://www.hydrology-and-earth-system-sciences.net/for_authors/manuscript_preparation.html

Regarding floods and ENSO reference should be given to Ward et al. (2010) and their findings should be mentioned. They find little influence of ENSO in the NE-US.

The number of tables is exaggerated and could for a large part be moved to a Supplement.

Page 16, 2$^{nd}$ bullet: This is a weak and scale-dependent conclusion. I could equally pick any number between 1 and infinite, assign that number of regions and calculate the percentage they contribute.

Page 16, 3$^{rd}$ bullet (and other bullets as well): Please direct the reader to the figures from which the conclusions are drawn. I cannot follow the reasoning around ENSO here.

**TECHNICAL CORRECTIONS**

It is kg and not Kg is often used in the paper.

The enormous amount of abbreviations makes the paper difficult to read. Please use abbreviations sparsely. The four source regions do not have to be abbreviated in my opinion. IWV is used only once, TNAO only twice, DFO only twice, thus it makes no sense to abbreviate. Please check for more of these examples. TME needs to be redefined in the summary section to be able for the cross-reader to read this sections stand-alone. Should it not be TMEs by the way? "N.E. USA" is supposed to be the abbreviation of Northeast United States as read in the abstract. As there is no "A" (see title) and the placement of the dots is a bit random I would suggest to write is as NE-US and it to be the abbreviation of the Northeastern United States.

P2:L18: "Tropical Moisture Exports (TME) was" → "were"

P4:L20: "enters" → "enter"

P5:L1: "mechanism" → mechanisms. There are simply too many plural/singular mistakes that I will not list any more.

P5:L1: "Atmospheric" → atmospheric

P5:L15: & → and

P11:L5: place a comma after First

**REFERENCES**

Dirmeyer, P. A., Brubaker, K. L. and DelSole, T.: Import and export of atmospheric water vapor between nations, J. Hydrol., 365(1–2), 11–22, doi:10.1016/j.jhydrol.2008.11.016, 2009.

Keys, P. W., van der Ent, R. J., Gordon, L. J., Hoff, H., Nikoli, R. and Savenije, H. H. G.: Analyzing precipitationsheds to understand the vulnerability of rainfall dependent regions, Biogeosciences, 9(2), 733–746, doi:10.5194/bg-9-733-2012, 2012.

Sprenger, M. and Wernli, H.: The LAGRANTO Lagrangian analysis tool – version 2.0, Geosci. Model Dev., 8, 2569-2586, doi:10.5194/gmd-8-2569-2015, 2015.

Ward, P. J., Beets, W., Bouwer, L. M., Aerts, J. C. J. H. and Renssen, H.: Sensitivity of river discharge to ENSO, Geophys. Res. Lett., 37(12), L12402, doi:10.1029/2010gl043215, 2010.

---

## Referee Comment (RC2) · Anonymous Referee #2 · 8 Oct 2016

Review of paper

"Tropical Moisture Exports, Extreme Precipitation and Floods in Northeast US"

by M. Lu and U. Lall

submitted to HESSD

This study investigates the role of tropical moisture exports (TMEs) for extreme precipitation and floods in the northeastern U.S. The authors use an previously published TME climatology by Knippertz and Wernli, and quantify the statistical linkage between TMEs entering the northeastern U.S. and precipitation and floods in the same region. The objective of this study is fine, but the quality of the text and figures is insufficient. Important aspects of the study are not well described, the text is confusing in several

places (and contains too many details), and again the figures provide too much information and lack clarity and compelling evidence. Since I think that a complete rewriting of the paper is required, including a redesign of the figures, I recommend very major revisions.

Major comments:

A) The writing should be strongly improved, for instance in the abstract:

- I am not sure how an air mass can be "born", and why "primarily"? The TME approach only considers air masses of tropical origin.

- "contribute to the global climatology precipitation and its extremes" –> "... of precipitation"

- I am not sure what is meant by "birth process and steering of TME"

- what is meant by "TME birth and entrance"? I assume that the authors mean "tropical origin of TMEs and where they reach the NE U.S."

- what is meant by an "extreme TME"?

B) Important meteorological aspects are not well described, in several places because sentence are too long and contain too much information, e.g.:

- p. 1 line 24 (the first sentence of the introduction): this sentence mixes too many things and become incorrect. The surface baroclinicity mainly drives the extratropical westerlies (jet stream), which then leads to Rossby waves, which can break, and these Rossby wave breaking events are likely involved in events of strong meridional moisture transport. Such meridional moisture transport occurs in extratropical cyclones, but then it typically has no tropical origin; in contrast TMEs (meridional transport with tropical origin) often occurs without extratropical cyclones.

- p. 3 line 9: "Lu et al. (2013) associated TME from the Gulf of Mexico and Tropical North Atlantic Ocean (TNAO) ..." –> no need to introduce an abbreviation if it is not used

later! "... east to the Bahamas islands" –> too much detail "... as the major moisture sources for the 1995 January flood in western France" –> end sentence here "... and demonstrated the predictability of the extreme precipitation given only the midlatitude sea level pressure (SLP) fields" –> totally different aspect, why mention predictability here? "...suggesting that steering mechanisms were important" –> isn't this trivial, of course the steering of moist air masses is important(?).

- p. 3 line 20: confusing mixture of "TME tracks" and "trajectories", why tracks? Please make clear that the TME approach is based on air parcel trajectories.

- p. 3 line 21: "Each trajectory has its moisture source calculated for every 100 km $\times$ 100 km box between the equator and $20°$N" –> what does this mean? The TME approach by Knippertz and Wernli does not calculate moisture sources! "... such that 90% of all water vapor is integrated" –> I don't understand this.

- p. 3 line 26: "To ensure that the characteristics of the tropical air parcels are maintained on their way across the subtropics" –> what is meant by this? Which characteristics should be "maintained"?

- p. 4 line 20: "The number of TME that enters the N.E. USA on any given day depends on the associated birth process" –> what is meant by this? Do you simply refer to the region of TME origin? Or do you speak about the processes that make the tropical air mass leave the tropics?

C) I don't see TMEs studies primarily as studies of moisture sources, TME studies mainly address the question "where when and how does tropical moisture reach the extratropics?" For specific moisture source studies, backward trajectories are more feasible and sophisticated techniques have been developed to obtain detailed moisture source fields for extreme precipitation and flood events, see, e.g., the following studies:

Sodemann, H., C. Schwierz, and H. Wernli, 2008. Inter-annual variability of Greenland winter precipitation sources: 1. Lagrangian moisture diagnostic and North Atlantic

Oscillation influence. J. Geophys. Res., 113, D03107, doi:10.1029/2007JD008503.

Winschall, A., S. Pfahl, H. Sodemann, and H. Wernli, 2014. Comparison of Eulerian and Lagrangian moisture source diagnostics – the flood event in eastern Europe in May 2010. Atmos. Chem. Phys., 14, 6605-6619.

Piaget, N., P. Froidevaux, P. Giannakaki, F. Gierth, O. Martius, M. Riemer, G. Wolf, and C. M. Grams, 2015. Dynamics of a local Alpine flooding event in October 2011: moisture source and large-scale circulation. Quart. J. Roy. Meteorol. Soc., 141, 1922-1937.

D) p. 4/5: very nice that the authors have specific research questions; they sound good, but they are unclear to the reader. What is an "entrance mechanism"? What are "identifiable atmospheric circulation patterns"? Why identifiable?

E) To me, the tables are much too detailed. Consider only showing seasonal mean values (not monthly) and maybe reduce the number of tables.

F) Figures:

- to me, Fig. 1 is not insightful, "month" and "ENSO" should not be similar categories

- Fig. 2: far too many panels, please reduce the information such that it becomes attractive for the reader

- Fig. 3: is this for all TMEs, or only for TMEs entering the northeastern U.S.?

- Fig. 3-5: my impression is that some information in these figures is redundant. Would Fig. 5 not summarize the key information and Figs. 3 and 4 could be omitted?

- Figs. 7 and 8: I think here the reader is completely lost, this information is not yet well "digested" by the authors. These monthly fields look so strikingly different such that I don't know what I can learn from these fields. Why showing the entire Northern Hemisphere? The key processes of TMEs entering the northeastern U.S. should be much more local. Also I have no idea what the unit is in these panels (500 Pa? = 5

hPa?). Why are the signals not much stonger near the U.S.?

- Fig. 10: caption is unclear.

Minor comments:

1) Title, should read "... in the northeastern U.S."

2) p. 2 line 17: instead of Wernli (1997), it was Knippertz and Martin (2007, Weather and Forecasting), who introduced the term "moisture conveyor belt". This sentence is not fully clear to me, I think the main difference between ARs and TMEs is the Eulerian vs. Lagrangian defintion - this should be better emphasized.

3) p. 3 line 6: "note" –> "noted"

4) p. 3 line 7: "tropical born moist air masses" –> again, "born" does not make sense; air masses cannot be born or die, what changes is their moisture content, and therefore we can speak about moisture sources and sinks

5) p. 5 line 14: unit should be "kg" not "Kg"

6) p. 5 line 14: "The position of the air parcel was updated every 6 hours, thus each track has 29 (4 updates up to 7 days including birth place, 4×7+1) positions (latitudes & longitudes) recorded on its trajectory." This is terribly complicated and not understandable, why not just "For all TME trajectories, position information (lon, lat) is available every 6 hours". On line 21 you say the same thing again.

7) p. 5 line 22: "death location" is very awkward, this is just the end of the TME trajectory calculation!
* * *

---

## Author Comment (AC1) · 18 Nov 2016

**REVIEW OF LU AND LALL**

**GENERAL COMMENTS**

This paper discusses the role of tropical moisture exports in extreme precipitation and flooding events in the Northeastern United States. The authors seem to have done a thorough investigation, but I found it difficult to assess the value of their results. My main points are:

Lack of comprehensiveness. I found it difficult to pick up the main messages from the paper, because it is lengthy with many figures and many abbreviations. Moreover, the text is often literally saying what is already seen in the figures, but not saying what we learn from these figures. The tables are often full of data, but could often be moved to a Supplement. The methods and definitions should be given more clearly. The conclusions are not 1-to-1 related to the research questions. I'd advice the authors to seriously condense their manuscript.

**Response: We appreciate the reviewer's comments and suggestion on improving the writing and organization of the results. We will address these points in our revised manuscript by (1) providing a more comprehensive discussion of the results with improved consistency with listed research questions and (2) reorganize the content substantially and shorten the manuscript.**

Context. What is the reason the Norteastern United States were picked as a study region? Is this part of a bigger project or something else? Why is ENSO given so much attention as previous studies showed little influence of ENSO in this region? These main point will also come back in the specific comments below.

**Response: The northeastern United States were chosen because we are located in this area and have experienced more and intense storms during the recent years. However, this area has not been widely studied in the context of tropical moisture transports associated with extreme storms. This area was also chosen since in a previous study for Western France, we saw moisture tracks for some extreme storms in France originating in the Gulf of Mexico and passing through N.E. US, with precipitation in the N.E. US enroute to France. A nearly planetary scale organization in atmospheric circulation marked the French flood event, that delivered far field moisture to France.**

**SPECIFIC COMMENTS**

1 and 1.1. The section 1.1 appears to be a mixture of literature review, objectives and method description. Please follow a more common outline. Part of it should go to the introduction and another part should be in Methods and merged with 2 Data, which is a misleading name as well, because it also discusses the modelling framework.

**Response: Thanks for the suggestion regarding the layout. We will improve on this in the revised manuscript. The equations provided in data section were for clarifying how we processed with the TME dataset, provided by Dr. Knippertz, and how we calculate the calendar day climatology. These are directly linked to the dataset we use, and it is not a modelling procedure.**

Why is not the latest version of the LAGRANTO model used (Sprenger and Wernli, 2015)?

**Response: The TME dataset was provided by Dr. Knippertz, who had previously run LAGRANTO and save the trajectories as a a comprehensive data base. Consequently, we did not run it again for our study.**

Is backward tracking from the NE- US applied or forward tracking and then integrated?

**Response: All tracks in the Knippertz database, that passed through the N.E. USA at any point were retained. The data base considers forward tracking and retains only those tracks that cross 15 deg N, and have a minimum duration, as described in the manuscript**

Is the LAGRANTO model run at all by the authors or did they post- process a dataset from previous work? This is all not clear to me.

**Response: Please see the response above, and the citation *Knippertz et al.* (2013) in the Section 1.1 Tropical Moisture Export Characterization.**

P2:L10-L11: There are many more regions in the world besides the NE-US that not have been studied much in this context. Is there any specific reason why the NE-US has been chosen as a region of study?

**Response: In addition to the consideration that this area has not yet been studied in**

this context, N.E. US is an interesting area that we observed seasonally changing spatial patterns of the TME tracks in our previous study (*Lu et al.*, 2013). **The trajectories have clear spatial patterns that are regulated by seasonal atmospheric circulations. And we have experienced more and intensive extreme events in the past several years, an improved understanding of the hydrometeorological factors associated with them has potential for contribution is needed.**

P2:L14-24: Some suggested definitions are discussed for ARs, "moist conveyor belt" and TME, but what is the exact definition that is used in this paper?

**Response: The focus of this study is the tropical moisture exports, which is a broader concept to study the movement of moist air and its linkage with precipitation and associated atmospheric drivers. The definitions provided in the manuscript are the commonly adapted definitions of ARs. They are given in the manuscript to serve as literature review and to show that the concepts of ARs limit a general study; on the other hand, the identification procedure of Tropical Moisture Exports (Knippertz and Wernli, 2010) consists three steps: (1) identify the tropical moisture sources between 0 and 20°N, between 1000 and 490 hPa; (2) retain tracks that reach 35°N within 6 days, this relatively short time span is also to ensure that the selected air parcels are likely to maintain their characteristics of tropical air when they cross the subtropics; (3) further select the tracks by only retaining those reach a water vapor flux of at least 100 k kg$^{-1}$ m s$^{-1}$ somewhere north of 35°N. More details about the definition can be found in Knippertz and Wernli (2010).**

P3:L23-25: "Each trajectory represents $3 * 10^{12}$ kg of atmospheric mass" I do not understand the (relevance of) this statement. How many parcels are released from the vertical? Why is the unit not kg m$^{-2}$?

**Response:. We clarify this in the revised manuscript as "One-day forward trajectories are calculated from every 100 km X 100 km X 30 hPa box between 0 and 20N, between 1000 hPa and 490 hPa. Specific humidity can be converted into water mass as each trajectory represents the same atmospheric mass of ~ 3 X 10$^{12}$ kg (Knippertz and Wernli, 2010).**

P4:L14: I suppose that little HESS readers will be familiar with the mei-yu-baiu front. Some explanation is required for readability. The four regions should also be outlined in a Figure. Why isn't sea level pressure data used from the same source as the input to LAGRANTO (ERA-Interim)? Sea level pressure and

Oceanic Niño Index is being used how and why exactly? Four paragraphs are used to discuss Fig. 2. Could this not be summarized in one table?

**Response: Since mei-yu-baiu front is not very relevant to the focus of this study, we didn't extend the discussion about this. We will consider the suggestion of providing a figure or a reference for the for TME hotspots regions. We have used NCEP/NCAR Reanalysis dataset before and the quality of the data is assessed and the data resolution and characteristics fit the objective and plan of this study.**

P9:L25-26: Is this conclusion drawn from Fig. 2? Or from somewhere else?

**Response: It is drawn from the entire study. The sentence here also provides a reason why we further analyse other factors in Fig.1 to complete the framework.**

P10: I'd expect a general conclusion from Fig. 3, besides the discussion of the individual panels only. The influence of ENSO seems rather small in general, and definitely not statistically significant, as the neutral years or often not in between El Niño and La Niña, am I right?

**Response: The influence of ENSO is seasonal, and we have provided a detailed explanation for each TME source region on which season, the separation between El Niño and La Niña is statistically significant. And the neutral ENSO state line is not necessarily to be always between El Niño and La Niña lines, because the lines show the average number of tracks both in the region. We are interested in the influence of abnormal ENSO years, i.e., El Niño or La Niña.**

7 Summary and Discussion: There appear to be 5 key findings, but it would help the reader if the 4 research questions from Page 4 are exactly answered. Moreover, this section could be named Conclusions and significantly condensed.

**Response: Thanks for the excellent suggestion. We will modify this section accordingly.**

Figure 1: What is the meaning of the different colors of the arrows? What does EP mean? None of it is explained in the caption.

**Response: EP, extreme precipitation, is introduced in P4 Line 23. Starting from P4 line 19, we introduced and explained Figure 1, the colors correspond to different levels of our analysis of the dependence. We will further clarify this in the manuscript by adding the explanation on the colors, added text in bold:** The conceptual framework of the analysis presented in this paper is indicated in Figure 1. The causal

structure illustrated considers the potential dependence of the TME Birth process as a function of the source location, the season and ENSO state **(red arrows in Figure 1)**. The number of TME that enters the N.E. USA on any given day depends on the associated birth process, the season, the source, the ENSO state, and the atmospheric circulation **(blue arrows in Figure 1)**. The total water released ($\Delta Q$) by the TME in the N.E. USA on a given day is taken to depend on the number of TME entering; the extreme precipitation amount, EP, is considered to depend on the $\Delta Q$ **(black arrows in Figure 1)**.

Figure 2: The panels are really tiny and difficult to read. I'd advice to split this into at least two figures or maintain only the most important panels. Please also provide this as a movie, pdf or ppt in the Supplement with each individual panel in one view.

**Response: We will take the reviewer's suggestion to provide a supplementary document containing each figure in a separate page.**

Figure 2: What is the exact definition of a storm track being born?

**Response: In the data section, we've stated that "…daily tracks born in the tropics that meet the following criteria: (1) they reach 35°N within the next 6 days after crossing 20°N, and (2) water vapor flux of any track is not less than 100 g kg$^{-1}$ m s$^{-1}$."**

Figure 2: I suppose the dots are connected to become actual tracks? This is not distinguishable from the tiny panels. Does all tracks "end" in the NE-US? If so, what is the exact definition of the "end" of a storm track?

**Response: A supplementary document will be provided. As stated in the manuscript data section that "… each track has 29 (4 updates up to 7 days including birth place, 4×7+1) positions (latitudes & longitudes) recorded on its trajectory." The end of a track is the position on the 7th day after it was born in the tropics. TME tracks can end anywhere, but when we analyze the TME entrance in section 5, we focus on tracks that entered the area, those tracks could enter the study region and exit.**

Figure 3: I suppose that the number of tracks is not really a physical quantity, but dependent on the resolution used in LAGRANTO or am I wrong? This should be explained.

**Response: It is not a physical quantity. More details regarding the dataset can be found in (Knippertz and Wernli, 2010), but we have emphasized in the manuscript in**

**section 1.1 that "To ensure that the characteristics of the tropical air parcels are maintained on their way across the subtropics, only trajectories that reach 35°N within the next 5 to 6 days after crossing 20°N were retained; … The water vapor fluxes of the retained tracks in the dataset must reach 100 g Kg-1 m s-1, a threshold chosen to represent 'fast' events and yet get meaningful statistics (Knippertz and Wernli, 2010)."**

Figure 3: Do all these TMEs also go to the NE-US? If not, what is the value of this figure?

**Response: The figure corresponds to the diagnosis of the seasonality and interannual variability of the moist tracks born in the tropics. It is based on all tracks born in these four regions. It is an important step to complete the framework in Fig.1.**

Figure 4: Is a birthplace (source region) the same as an evaporative source (e.g. Keys et al., 2012) or something else?

**Response: Yes, the birthplace of a track corresponds to where the convection occurred.**

Figure 5b: This is very much scale-dependent and that should be mentioned.

**Response: Thanks – we will clarify this**

Figure 6: P stands for what? Precipitation, probability? I suppose probability, but it is not defined. As the ENSO signal does not do much, could this whole figure not be merged with Fig. 5b to show the variability?

**Response: It has been stated in the manuscript that the formulas, P(X|Y) correspond to conditional probability. We will include a reference on the how we calculate the conditional probability in this study. It actually follows the classical calculation. Figure 5(b) focuses on comparison among all sources; while Figure 6 focuses on separation under different ENSO states. We will consider the reviewer's suggestion to find a better way, also condensed, to present the results.**

Results could be compared to http://cola.gmu.edu/wcr/ (Dirmeyer et al., 2009) and then looking at the St. Lawrence river basin.

**Response: We thank the reviewer's suggestion on reference, and we will incorporate into revised manuscript.**

I am missing the entire point of Figures 7 and 8. What do they explain?

**Response: Figure 7 and 8 link the atmospheric circulation with TME entrance. It compares the composite anomalies between active and inactive TME entrance days. For the TME to enter the study area, the associated atmospheric circulation pattern has to be in favour of such convergence of moist air.**

Figure 9 and Page 14 (and other places as well): Precipitation is a flux is should be defined per unit of time. The time integrator of $\Delta Q$ is not given. See http://www.hydrology-and-earth-system-sciences.net/for_authors/manuscript_preparation.html

**Response: In Eqn. (2), the change of total specific humidity integrated over all the tracks is calculated for a given date. Our analysis refers to this enter date and focuses on the total precipitation due to TME. We will clarify this in the revised manuscript**

Regarding floods and ENSO reference should be given to Ward et al. (2010) and their findings should be mentioned. They find little influence of ENSO in the NE-US.

**Response: We thank the reviewer's suggestion on reference, and we will incorporate into revised manuscript. It might be worth mentioning here again that our results suggest a nonlinear relationship between ENSO and TME in N.E. US. This is quite different than the analysis in Ward et al. (2010) for ENSO correlations with computed seasonal discharge.**

The number of tables is exaggerated and could for a large part be moved to a Supplement.

**Response: Thanks. We will reorganize the tables and move to supplementary document.**

Page 16, 2[nd] bullet: This is a weak and scale-dependent conclusion. I could equally pick any number between 1 and infinite, assign that number of regions and calculate the percentage they contribute.

**Response: We will clarity this point further in the revised manuscript.**

Page 16, 3[rd] bullet (and other bullets as well): Please direct the reader to the figures from which the conclusions are drawn. I cannot follow the reasoning around ENSO here.

**Response: We will clarity this point further in the revised manuscript.**

**TECHNICAL CORRECTIONS**

It is kg and not Kg is often used in the paper.

**Response: Typo is corrected.**

The enormous amount of abbreviations makes the paper difficult to read. Please use abbreviations sparsely. The four source regions do not have to be abbreviated in my opinion. IWV is used only once, TNAO only twice, DFO only twice, thus it makes no sense to abbreviate. Please check for more of these examples. TME needs to be redefined in the summary section to be able for the cross-reader to read this sections stand-alone. Should it not be TMEs by the way? "N.E. USA" is supposed to be the abbreviation of Northeast United States as read in the abstract. As there is no "A" (see title) and the placement of the dots is a bit random I would suggest to write is as NE-US and it to be the abbreviation of the Northeastern United States.

**Response: We really appreciate the reviewer's suggestion on this writing details. We will incorporate all these in the revised manuscript.**

**Response: All the following typos are corrected.**

P2:L18: "Tropical Moisture Exports (TME) was" "were"

P4:L20: "enters" "enter"

P5:L1: "mechanism" mechanisms. There are simply too many plural/singular mistakes that I will not list any more.

P5:L1: "Atmospheric" atmospheric

P5:L15: & and

P11:L5: place a comma after First

**REFERENCES**

Dirmeyer, P. A., Brubaker, K. L. and DelSole, T.: Import and export of atmospheric water vapor between nations, J. Hydrol., 365(1–2), 11–22, doi:10.1016/j.jhydrol.2008.11.016, 2009.

Keys, P. W., van der Ent, R. J., Gordon, L. J., Hoff, H., Nikoli, R. and Savenije, H. H. G.: Analyzing precipitationsheds to understand the vulnerability of rainfall dependent regions, Biogeosciences, 9(2), 733–746, doi:10.5194/bg-9-733-2012, 2012.

Sprenger, M. and Wernli, H.: The LAGRANTO Lagrangian analysis tool – version 2.0, Geosci. Model Dev., 8, 2569-2586, doi:10.5194/gmd-8-2569-2015, 2015.

Ward, P. J., Beets, W., Bouwer, L. M., Aerts, J. C. J. H. and Renssen, H.: Sensitivity of river discharge to ENSO, Geophys. Res. Lett., 37(12), L12402, doi:10.1029/2010gl043215, 2010.

---

## Author Comment (AC2) · 18 Nov 2016

Review of paper  "Tropical Moisture Exports, Extreme Precipitation and Floods in Northeast US" by M. Lu and U. Lall  submitted to HESSD

This study investigates the role of tropical moisture exports (TMEs) for extreme precipitation and floods in the northeastern U.S. The authors use an previously published TME climatology by Knippertz and Wernli, and quantify the statistical linkage between TMEs entering the northeastern U.S. and precipitation and floods in the same region. The objective of this study is fine, but the quality of the text and figures is insufficient. Important aspects of the study are not well described, the text is confusing in several places (and contains too many details), and again the figures provide too much information and lack clarity and compelling evidence. Since I think that a complete rewriting of the paper is required, including a redesign of the figures, I recommend very major revisions.

Major comments:

A) The writing should be strongly improved, for instance in the abstract:

- I am not sure how an air mass can be "born", and why "primarily"? The TME approach only considers air masses of tropical origin.

**Response: We thank the reviewer for the questions. The birth and death of TME refer to the starting and end points of the tracks**

- "contribute to the global climatology precipitation and its extremes" –> "... of precipitation"

**Response: We think it is redundant to add "… of precipitation" here.**

- I am not sure what is meant by "birth process and steering of TME"

**Response: They refer to its formation in the tropics and its movement, respectively.**

- what is meant by "TME birth and entrance"? I assume that the authors mean "tropical origin of TMEs and where they reach the NE U.S."

**Response: Yes.**

- what is meant by an "extreme TME"?

**Response: Intensive TME activities.**

B) Important meteorological aspects are not well described, in several places because sentence are too long and contain too much information, e.g.:

**Response: We appreciate the reviewer's comments here and will take these into consideration for the revision of the manuscript.**

- p. 1 line 24 (the first sentence of the introduction): this sentence mixes too many things and become incorrect. The surface baroclinicity mainly drives the extratropical westerlies (jet stream), which then leads to Rossby waves, which can break, and these Rossby wave breaking events are likely involved in events of strong meridional moisture transport. Such meridional moisture transport occurs in extratropical cyclones, but then it typically has no tropical origin; in contrast TMEs (meridional transport with tropical origin) often occurs without extratropical cyclones.

**Response: Thanks. We will rewrite this to make it clear**

- p. 3 line 9: "Lu et al. (2013) associated TME from the Gulf of Mexico and Tropical North Atlantic Ocean (TNAO) ..." –> no need to introduce an abbreviation if it is not used later! "... east to the Bahamas islands" –> too much detail "... as the major moisture sources for the 1995 January flood in western France" –> end sentence here "... and demonstrated the predictability of the extreme precipitation given only the midlatitude sea level pressure (SLP) fields" –> totally different aspect, why mention predictability here? "...suggesting that steering mechanisms were important" –> isn't this trivial, of course the steering of moist air masses is important(?).

**Response: We will revise this given the comment. Yes, predictability is a separate point and we will develop it as such. A purpose of understanding the associated mechanism is to see if it offers an opportunity for prediction beyond the usual time scale of average predictability. Steering of moist air masses is important and in this paper and the cited paper it is shown as a determinant of the extreme precipitation. However, in presentations and in comments to the Lu (2013) paper several hydrologists asserted that local moisture recycling and/or land based sources may be a dominant precipitation mechanism. For the extreme precipitation events, we find that the TMEs may actually play a significant role.**

- p. 3 line 20: confusing mixture of "TME tracks" and "trajectories", why tracks? Please

make clear that the TME approach is based on air parcel trajectories.

**Response: We'll make it clearer in the revision, and use one term consistently**

- p. 3 line 21: "Each trajectory has its moisture source calculated for every 100 km × 100 km box between the equator and 20°N" –> what does this mean? The TME approach by Knippertz and Wernli does not calculate moisture sources! "... such that 90% of all water vapor is integrated" –> I don't understand this.

**Response: We will clarify this in the revised version and provide a stepwise procedure used.**

- p. 3 line 26: "To ensure that the characteristics of the tropical air parcels are maintained on their way across the subtropics" –> what is meant by this? Which characteristics should be "maintained"?

**Response: We will clarify this in the revised version and reproduce the criteria from Knippertz and Wernli (2010).**

- p. 4 line 20: "The number of TME that enters the N.E. USA on any given day depends on the associated birth process" –> what is meant by this? Do you simply refer to the region of TME origin? Or do you speak about the processes that make the tropical air mass leave the tropics?

**Response: The birth process here refers to where and how many tracks are born, how many were born, and how these vary by season. We will restate this to say exactly that**

C) I don't see TMEs studies primarily as studies of moisture sources, TME studies mainly address the question "where when and how does tropical moisture reach the extratropics?" For specific moisture source studies, backward trajectories are more feasible and sophisticated techniques have been developed to obtain detailed moisture source fields for extreme precipitation and flood events, see, e.g., the following studies:

Sodemann, H., C. Schwierz, and H. Wernli, 2008. Inter-annual variability of Greenland winter precipitation sources: 1. Lagrangian moisture diagnostic and North Atlantic Oscillation influence. J. Geophys. Res., 113, D03107, doi:10.1029/2007JD008503.

Winschall, A., S. Pfahl, H. Sodemann, and H. Wernli, 2014. Comparison of Eulerian and Lagrangian moisture source diagnostics – the flood event in eastern Europe in May 2010. Atmos. Chem. Phys., 14, 6605-6619.

Piaget, N., P. Froidevaux, P. Giannakaki, F. Gierth, O. Martius, M. Riemer, G. Wolf, and C. M. Grams, 2015. Dynamics of a local Alpine flooding event in October 2011: moisture source and large-scale circulation. Quart. J. Roy. Meteorol. Soc., 141, 1922- 1937.

**Response: You are right as to the point of TME studies. Definitely, as indicated in the references cited, a backwards trajectory approach for each event will identify the associated moisture sources. Since the data base available from Knippertz is based on a forward trajectory analysis, we censored those data to retain only the tracks that pass through our area of interest. As a result, we are not able to identify all moisture sources associated with the events, and are only identifying the TME contribution to the precipitation associated with the event. We have used HYSPLIT in other work for backwards trajectory identification, and indeed at the event scale that is the way to identify the fraction from each potential source. Here, we were trying to take advantage of the pre-processed data from Knippertz, and identify just how the tropical sources varied by space and season in their influence on extreme precipitation events in our region, using this data set.**

D) p. 4/5: very nice that the authors have specific research questions; they sound good, but they are unclear to the reader. What is an "entrance mechanism"? What are "identifiable atmospheric circulation patterns"? Why identifiable?

**Response: We'll revise the language here to be clearer.**

E) To me, the tables are much too detailed. Consider only showing seasonal mean values (not monthly) and maybe reduce the number of tables.

**Response: We'll reorganize the tables and figures in a way that total number of tables and figures will be reduced and some will be moved to supplementary document.**

F) Figures:  - to me, Fig. 1 is not insightful, "month" and "ENSO" should not be similar categories

**Response: We were considering these as factors influencing TME occurrence – namely seasonality and ENSO – can separate.**

- Fig. 2: far too many panels, please reduce the information such that it becomes attractive for the reader

**Response: Will simplify and separate to supplement**

- Fig. 3: is this for all TMEs, or only for TMEs entering the northeastern U.S.?

**Response: All TMEs born in the four regions respectively.**

- Fig. 3-5: my impression is that some information in these figures is redundant. Would Fig. 5 not summarize the key information and Figs. 3 and 4 could be omitted?

**Response: We'll reorganize the tables and figures in a way that total number of tables and figures will be reduced and some will be moved to supplementary document.**

- Figs. 7 and 8: I think here the reader is completely lost, this information is not yet well "digested" by the authors. These monthly fields look so strikingly different such that I don't know what I can learn from these fields. Why showing the entire Northern Hemisphere? The key processes of TMEs entering the northeastern U.S. should be much more local. Also I have no idea what the unit is in these panels (500 Pa? = 5 hPa?). Why are the signals not much stonger near the U.S.? - Fig. 10: caption is unclear.  Minor comments:  1) Title, should read "... in the northeastern U.S."

**Response: Figure 7 and 8 link the large scale atmospheric circulation with TME entrance. They compare  the composite anomalies between active and inactive TME entrance days. For the TME to enter the study area, the associated atmospheric circulation pattern has to be in favour of such convergence of moist air, and we think the large-scale organization contributes to the local organization as shown in our previous study Lu et al., (2013) in Western France, and it also relates to the wave interaction hypothesis advanced in Screen and Simmonds  2014. The unit is added in the revision and the title is modified accordingly.**

Screen, J. A., & Simmonds, I. (2014). Amplified mid-latitude planetary waves favour particular regional weather extremes. *Nature Climate Change*, *4*(8), 704-709.

2) p. 2 line 17: instead of Wernli (1997), it was Knippertz and Martin (2007, Weather and Forecasting), who introduced the term "moisture conveyor belt". This sentence is not fully clear to me, I think the main difference between ARs and TMEs is the Eulerian vs. Lagrangian defintion - this should be better emphasized.

**Response: We appreciate the reviewer's comment on this. It is very helpful, we will incorporate this in the literature review part.**

3) p. 3 line 6: "note" –> "noted"

**Response: Typo is corrected.**

4) p. 3 line 7: "tropical born moist air masses" –> again, "born" does not make sense; air masses cannot be born or die, what changes is their moisture content, and therefore we

can speak about moisture sources and sinks

**Response: The birth and death corresponds to the fact that the data only records up to 7 days. We want to avoid misleading to the commonly adapted definition of sources and sinks.**

5) p. 5 line 14: unit should be "kg" not "Kg"

**Response: Typo is corrected.**

6) p. 5 line 14: "The position of the air parcel was updated every 6 hours, thus each track has 29 (4 updates up to 7 days including birth place, 4×7+1) positions (latitudes & longitudes) recorded on its trajectory." This is terribly complicated and not understandable, why not just "For all TME trajectories, position information (lon, lat) is available every 6 hours". On line 21 you say the same thing again.

**Response: We will rewrite as suggested.**

7) p. 5 line 22: "death location" is very awkward, this is just the end of the TME trajectory calculation!

**Response: Yes.**